# Drought alters the biogeochemistry of boreal stream networks

Lluís Gómez-Gener[1,2 ✉], Anna Lupon[3], Hjalmar Laudon [4] & Ryan A. Sponseller[1]

Drought is a global phenomenon, with widespread implications for freshwater ecosystems. While droughts receive much attention at lower latitudes, their effects on northern river networks remain unstudied. We combine a reach-scale manipulation experiment, observations during the extreme 2018 drought, and historical monitoring data to examine the impact of drought in northern boreal streams. Increased water residence time during drought promoted reductions in aerobic metabolism and increased concentrations of reduced solutes in both stream and hyporheic water. Likewise, data during the 2018 drought revealed widespread hypoxic conditions and shifts towards anaerobic metabolism, especially in headwaters. Finally, long-term data confirmed that past summer droughts have led to similar metabolic alterations. Our results highlight the potential for drought to promote biogeochemical shifts that trigger poor water quality conditions in boreal streams. Given projected increases in hydrological extremes at northern latitudes, the consequences of drought for the health of running waters warrant attention.

[1] Department of Ecology and Environmental Science, Umeå University, Linnaeus väg 6, 90736 Umeå, Sweden. [2] Stream Biofilm and Ecosystem Research Laboratory, School of Architecture, Civil and Environmental Engineering, Ecole Polytechnique Fédérale de Lausanne, GR A0 412 (Bâtiment GR), CH-1015 Lausanne, Switzerland. [3] Integrative Freshwater Ecology Group, Center for Advanced Studies of Blanes (CEAB-CSIC), 17300 Blanes, Girona, Spain. [4] Department of Forest Ecology and Management, Swedish University of Agricultural Sciences (SLU), 90183 Umeå, Sweden. ✉email: gomez.gener87@gmail.com

D roughts are among the most dramatic climate change impacts to the biosphere, resulting in severe ecological and socioeconomic costs at local, regional, and global scales[1–3]. These events originate when persistent atmospheric anomalies trigger below-normal soil moisture and propagate through the hydrological cycle to ultimately cause low or zero-flow conditions in river networks[4], also known as hydrological droughts. These events are intensified by anthropogenic activities, both directly through surface or groundwater abstractions, water diversions, and dam constructions, and indirectly as a consequence of land-use changes[2,5]. Hydrological droughts are common in regions with arid and semiarid climates, where the consequences for aquatic ecosystems are well documented[6,7]. Yet current models predict an increased occurrence and intensity of drought in regions where such events have been less common historically[8,9] and where their effects on freshwater resources are largely unstudied[10,11]. This includes high-latitude regions (north of ~55°N), which comprise ~33% of the global river network[12] and drain the world's most extensive soil carbon (C) reserves[13]. Indeed, the abundant headwater streams in northern landscapes play key roles as recipients and processors of terrestrial dissolved organic matter (DOM)[14], sources of greenhouse gas (GHG) emissions[15], habitat for aquatic biota[16], and regulators of downstream water chemistry[17]. Thus, how these ecosystems respond to drought may have far-reaching environmental and socioeconomic consequences for high-latitude regions.

Hydrological droughts affect stream water chemistry through a number of mechanisms[18], including fundamental shifts in metabolic processes that underpin biogeochemical cycles[19]. The diversity and rates of metabolic processes are determined by the co-occurrence of electron donors (e.g., organic substrates) and acceptors (e.g., dissolved oxygen, nitrate, iron, manganese, sulfate, and carbon dioxide), the latter of which are used in descending order of the energy generated by their reduction[20]. Drought can act upon these processes by reducing the hydrological transport of organic substrates from soils to streams[21], by increasing the water residence time (WRT) during which different electron acceptors may be used[22], and by restricting the resupply of dissolved oxygen ($O_2$)[23], the most energetically favorable electron acceptor, through decreased water–atmosphere gas exchange. Consequently, as drought ensues, localized $O_2$ depletion can lead to a wide range of anaerobic microbial processes, including methanogenesis, the least energetically favorable pathway[20]. While these effects are general, northern streams may be particularly prone to such biogeochemical shifts during drought because extensive organic matter storage at the land–water interface[24] promotes reducing conditions in near-stream environments, as well as relatively high and persistent supply of DOM[25]. Hence, in these systems, even small increases in WRT and the associated reductions in gas exchange may cause a depletion in $O_2$ and increased rates of anaerobic processes in underlying sediments. Such low-flow events, depending on their frequency, intensity, and duration, are likely to alter the roles that streams play as processors of C and nutrients, as well as habitat for aquatic communities.

Here, we tested the hypothesis that periods of drought alter water chemistry in high-latitude streams by shifting the metabolic pathways by which DOM is processed under different redox conditions. We also evaluated to what degree metabolic responses to drought influence network-scale biogeochemical patterns, and if these have implications for the water quality of northern streams. To test these effects, we performed a reach-scale drought manipulation experiment (in 2017) along a 1.4-km boreal headwater stream in the Krycklan Catchment Study (KCS), located in northern Sweden. We complemented experimental results at the reach scale by exploring network-scale responses to a major

drought that hit Northern Europe the following summer (in 2018). This was one of the most severe droughts documented in this region over the last 100 years (Fig. 1a)[26], having widespread effects on catchment hydrology, with many streams in the region experiencing record-low flows (Fig. 1b, c)[27]. Finally, we explored stream biogeochemical responses to past drought periods through analysis of historical data from a set of headwater catchments within the KCS monitoring program.

## Results and discussion

**Reach-scale responses to experimental drought.** We experimentally simulated drought over a 2-week period during summer 2017 by damming a lake outlet that feeds a small stream in the KCS (Supplementary Fig. 1). The broader KCS landscape is typical of boreal Fennoscandia, dominated by coniferous forests (*Pinus sylvestris* and *Picea abies*), open wetlands (mires) with extensive peat accumulation, and several headwater lakes[28] (Supplementary Table 1). Upland soils are primarily well-developed iron podzols, but thicker, organic-rich deposits are common in the riparian zones of headwater streams[29]. The experimental stream reach drains a headwater catchment (1.1 ha) covered by a mix of coniferous forests (71%) and open mires (25%) that surround a small lake (C6 in Supplementary Fig. 1a and Supplementary Table 1). Flow manipulation reduced the average discharge among six 50-m study segments (Supplementary Fig. 1b, Supplementary Table 2) from 12.3 to 1.1 L s$^{-1}$, which translated to a prolongation of local WRT from 28.0 to 223.1 min (Supplementary Fig. 2). However, our experiment did not simulate drought on land, and thus lateral groundwater inflows along the reach were initially sustained, supplying water, solutes, and gases at the onset of the experiment (Supplementary Fig. 3). The change in water source when the lake was dammed led to a small decrease in the average daily water temperature along the reach, from 12.7 ± 1.8 to 10.4 ± 1.1 °C (mean ± SD). Importantly, lateral hydrologic inputs were patchy[30], and thus generated a gradient in drought severity, such that WRT varied locally among segments from 14.3 to 1061.3 min (Supplementary Fig. 2b). Finally, the strength of these lateral connections declined as the experiment progressed and WRT increased, and eventually the water table dropped to the point that we could no longer draw water from near-stream wells installed at 0.5–1-m depth (Supplementary Fig. 3).

Within days of inducing drought, we observed a reduction of $O_2$ in both the surface and hyporheic water of the experimental stream (Fig. 2a). In both cases, $O_2$ concentration decreased nonlinearly with greater WRT, highlighting the hydrological dependency of the vertical, lateral, and longitudinal vectors of $O_2$ transport and atmospheric exchange in streams[23]. However, this reduction was more abrupt and persistent in the hyporheic zone, where all observations remained below the critical saturation level of 25% once WRT surpassed 200 min. Independent estimates of aerobic metabolism (Methods and Supplementary Methods 1) mirrored patterns of $O_2$ concentrations observed in hyporheic water during the experiment (Fig. 2b). Specifically, drought caused a significant reduction in aerobic respiration along the experimental stream reach, which decreased from −403 ± 172 to −130 ± 81 mmol $O_2$ m$^{-2}$ d$^{-1}$. Previous studies testing the effects of drought, either experimentally or under natural conditions, have shown that low flows can either enhance[31,32] or reduce[33] rates of stream aerobic respiration. In our case, aerobic respiration decreased nonlinearly with WRT ($r^2 = 0.41$; $p < 0.001$; $n = 111$), ostensibly because the biochemical $O_2$ demand driven by aerobic respiration greatly exceeded the resupply of $O_2$ to hyporheic sediments as drought ensued.

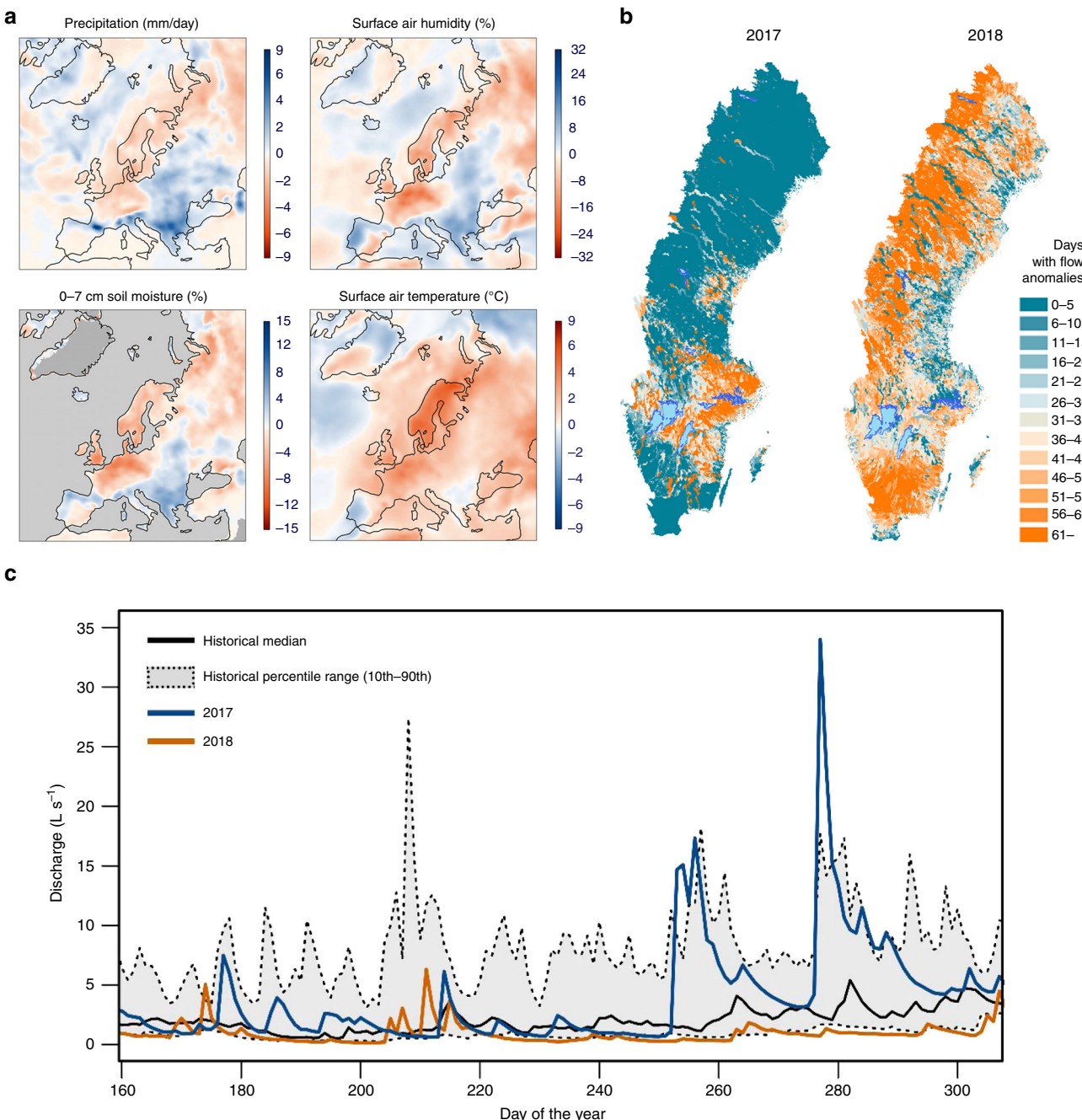

**Fig. 1 The summer 2018 drought in northern Europe. a** Spatial distribution of July 2018 anomalies of the primary factors controlling the water balance of watersheds over Europe (average deviation for July 2018 relative to the monthly average for the period 1979–2018; Source: European Centre for Medium-Range Weather Forecasts (ECMWF), Copernicus Climate Change Service (C3S)). **b** Comparison of the spatial distribution of summer flow anomalies for 2017 and 2018 over Sweden (daily flow deviation for summer 2017 and 2018 relative to the daily summer flows for the period 1963–1992; Source: Swedish Meteorological and Hydrological Institute (SMHI)). **c** Summer 2018 streamflow anomaly for a headwater stream in the KCS. The figure shows the median daily discharge and the daily 10th to 90th percentile values during summer between 1985 and 2018 (black solid line and gray shade, respectively). Daily discharge for 2017 and 2018 periods is shown in blue and orange solid lines, respectively.

Concurrent to reductions in $O_2$ concentration and aerobic metabolism, reduced forms of redox-sensitive solutes and gases also accumulated in the stream as experimental drying progressed, an observation consistent with thermodynamic principles[20]. The influence of increasing WRT on this overall chemical change was clear from a principal component analysis (PCA) based on multiple solutes, and particularly for hyporheic waters (Supplementary Fig. 4). Similar patterns emerged for specific ratios of reduced to oxidized chemical forms (Fig. 3). For

example, increases in $NH_4^+$:$NO_3^-$ as drought progressed (Fig. 3a) were consistent with redox-driven changes in nitrogen cycling, including constraints on nitrification and upregulated rates of denitrification under low oxygen conditions[34]. Likewise, elevated $CH_4$:$O_2$ suggested an increasing transition through the full range of terminal electron-accepting processes, including methanogenesis, with greater WRT (Fig. 3b). Yet, during early stages of the experiment, these chemical ratios in the stream were also influenced by lateral groundwater inputs (Fig. 3; Supplementary

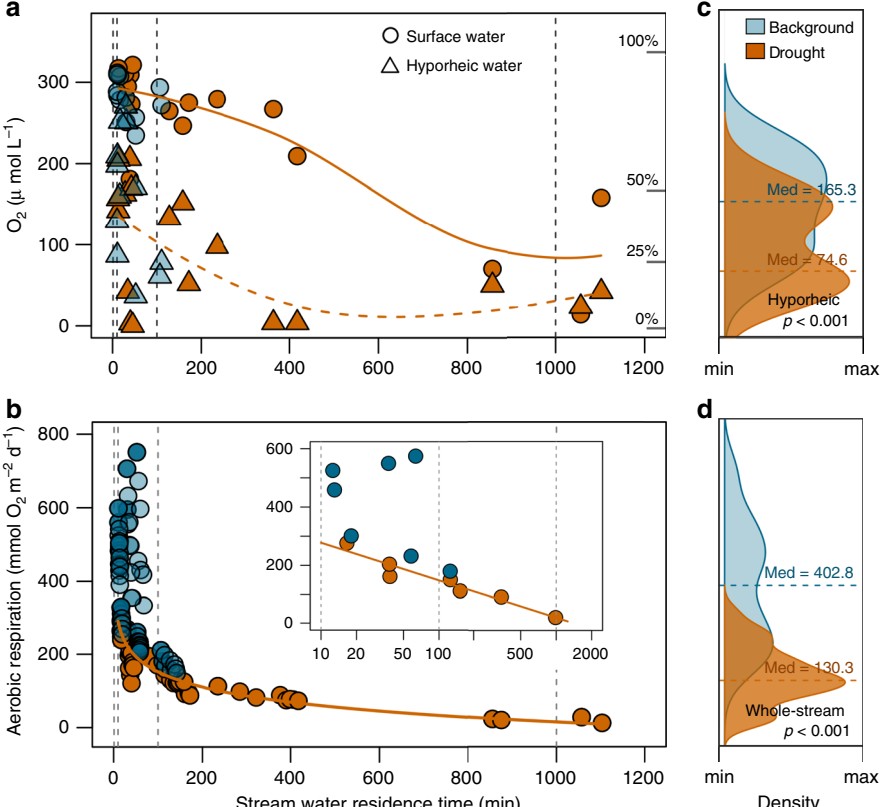

**Fig. 2 Experimental drought reduced O₂ availability and aerobic respiration.** Relationship between stream water residence time and **a**, dissolved oxygen ($O_2$) concentration in the stream surface and hyporheic water and **b**, aerobic respiration measured during the summer 2017 drought experiment. Panels **c** and **d** present Kernel density plots for hyporheic $O_2$ concentration and whole-stream aerobic respiration rates for drought and background (pre- and post drought) observations, respectively. Differences between drought and background conditions were tested using a nonparametric Wilcoxon Signed-Rank test. Orange and blue colors denote drought and background conditions, respectively. Circles and triangles denote surface and hyporheic water observations, respectively. Solid and dashed lines in panel **a** are locally weighted regression model fittings (Loess) for surface and hyporheic water observations, respectively. Solid orange lines in **b** represent the regression model best fitting the observations ($r^2 = 0.41$; $p < 0.001$, $n = 111$). The inset plot in panel **b** shows the segment-averaged aerobic respiration rates along a log-transformed $x$ axis.

Fig. 3), as well as by natural variation in the redox state of hyporheic sediments along the study reach (Fig. 3). This hydrological effect was also evident from the direct comparison of $CH_4$:$O_2$ at the same locations between the experimental (2017) and natural (2018) drought, which suggests qualitatively similar relationships with WRT, but relatively greater lateral inputs of $CH_4$ to the stream at low WRT during the experiment (Supplementary Fig. 5). $CH_4$:$O_2$ ratios ultimately converged as WRT increased beyond ca. 1000 min, and lateral groundwater inputs declined (Supplementary Fig. 3), suggesting that these chemical signals can be sustained by processes occurring within the stream ecosystem boundaries. At the same time, the differences between these two curves across the full range of WRT conditions illustrate how variation in local groundwater hydrology during the onset of drought can potentially exacerbate transitions toward reducing chemical conditions in the stream. Such influences are likely to be pronounced in boreal headwaters, where riparian soils are often peat rich and strongly anoxic environments[25].

The observed chemical patterns suggest that drought in boreal streams can induce shifts in the relative dominance of aerobic versus anaerobic metabolic processes. While measuring whole ecosystem rates of anaerobic metabolism in streams remains a challenge[35], one way to explore the relative significance of these processes is to evaluate the departures of $CO_2$ and $O_2$ from atmospheric equilibrium ($\Delta CO_2$:$\Delta O_2$)[36–38]. Briefly, theory

predicts that $CO_2$ and $O_2$ should inversely covary if aerobic mineralization of organic matter dominates the flux of both gases, and deviations from this relationship can reveal $CO_2$ production through anaerobic processes[36,37]. For example, the low dispersion and high proximity to the 1:−1 line of $\Delta CO_2$:$\Delta O_2$ from discrete samples collected during background conditions (Fig. 4a) are consistent with aerobic metabolism driving the coupled dynamics of $O_2$ and $CO_2$, with only two hyporheic samples deviating substantially from theoretical values. By comparison, experimental drought promoted higher $\Delta CO_2$:$\Delta O_2$, measured as the centroid from discrete observations, as well as by the linear slope (Fig. 4a; Supplementary Table 3), suggesting persistent $CO_2$ production via non-aerobic pathways. Similarly, high-frequency data from paired $O_2$ and $CO_2$ sensors in the surface stream revealed a shift toward higher $\Delta CO_2$:$\Delta O_2$ domains as WRT increased throughout the experiment (Fig. 4b)[36]. Finally, the total dispersion around the $\Delta CO_2$:$\Delta O_2$ relationship for discrete samples also increased in both the stream surface and hyporheic water during drought (Fig. 4a; Supplementary Table 3), consistent with an overall diversification of metabolic processes for the study segments more exposed to drying[39]. Taken together, the reduction of aerobic respiration, accumulation of reduced compounds, and $\Delta CO_2$:$\Delta O_2$ imbalances all point to the co-occurrence of diverse metabolic pathways[39], including methano-genesis[40], in response to increasing WRT. The fact that such changes were observed over a relatively short experimental period

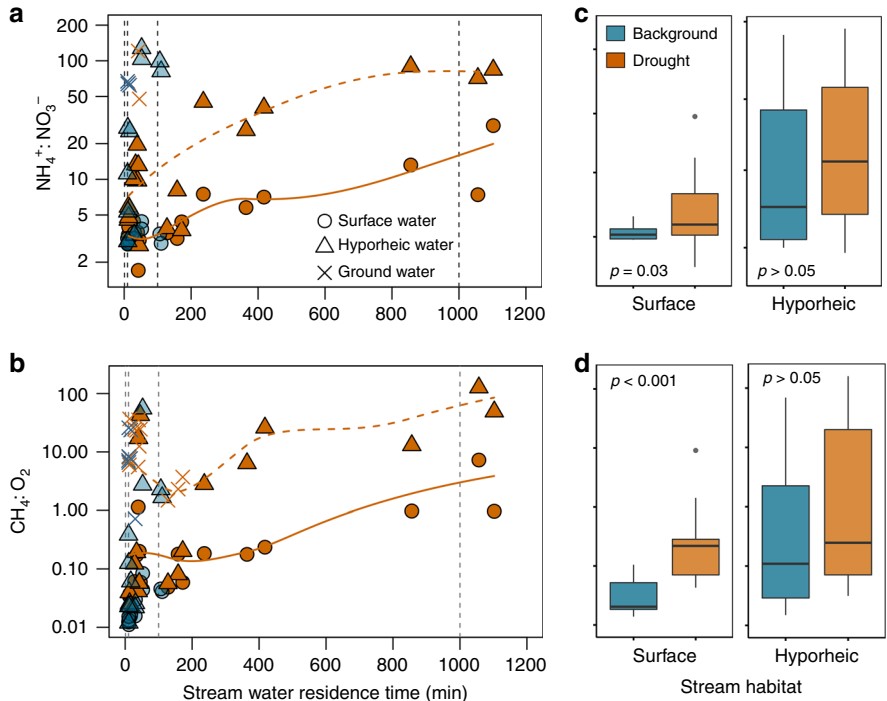

**Fig. 3 Experimental drought influenced NH$_4^+$:NO$_3^-$ and CH$_4$:CO$_2$ ratios.** Relationships between the stream water residence time and the molar ratios of **a**, NH$_4^+$:NO$_3^-$ and **b**, CH$_4$:O$_2$. Orange and blue colors denote drought and background (pre- and post drought) conditions, respectively. Circles and triangles denote surface and hyporheic water observations, respectively. Solid and dashed lines are locally weighted regression model fittings (Loess) for surface and hyporheic water observations, respectively. Note that crosses denote observations for groundwater well samples. Panels **c** and **d** show differences in NH$_4^+$:NO$_3^-$ and CH$_4$:O$_2$ ratios between the surface and hyporheic water, respectively. Box plots display the 25th, 50th, and 75th percentiles; whiskers display minimum and maximum values. Differences between hydrological conditions were tested using a nonparametric Wilcoxon Signed-Rank test.

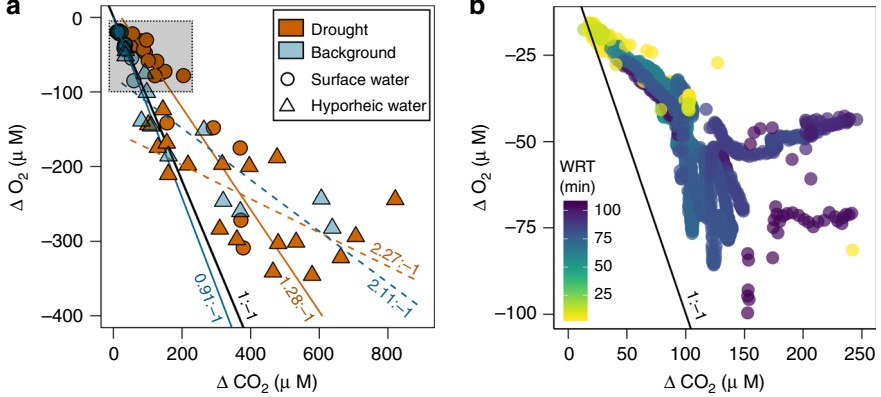

**Fig. 4 Experimental drought caused stochiometric imbalances between O$_2$ and CO$_2$.** Relationships between the molar departure of stream CO$_2$ and O$_2$ from atmospheric equilibrium ($\Delta$CO$_2$:$\Delta$O$_2$) assessed from **a**, low-frequency (i.e., grab samples) and **b**, high-frequency (i.e., sensor data) sampling. Circles and triangles denote surface and hyporheic water observations, respectively (note that panel **b** is only surface water). Orange and blue colors in panel **a** denote drought and background (pre- and post drought) hydrological conditions, respectively. Color pattern in panel **b** indicates changes in stream water residence time (WRT) during the experiment. The black line in panels **a** and **b** denotes the theoretical 1:−1 relationship defined for aerobic respiration. Solid and dashed lines in panel **a** represent the linear regression model for surface and hyporheic observations, respectively, either during drought (orange) or background (blue) conditions. The upper left square in panel **a** is a reference showing the space of the x–y plane captured by high-frequency data in panel **b**. Additional metrics associated with the analysis of $\Delta$CO$_2$:$\Delta$O$_2$ are presented in Supplementary Table 3.

(2 weeks) underscores the sensitivity of these headwater ecosystems to extreme low-flow conditions during drought.

**Network-scale biogeochemical responses to drought.** Summer 2018 provided a unique opportunity to explore how severe drought influences boreal stream chemistry at network scales.

This event was associated with extremely low summer discharge (Fig. 1b, c), as well as declines in dissolved organic C in streams (Supplementary Fig. 6), suggesting that channels had become isolated from lateral hydrologic connections to organic-rich soils[41,42]. Analysis of high-frequency O$_2$ data from 16 streams sites across the KCS (Supplementary Fig. 1a) revealed widespread network-scale reductions in O$_2$ concentration

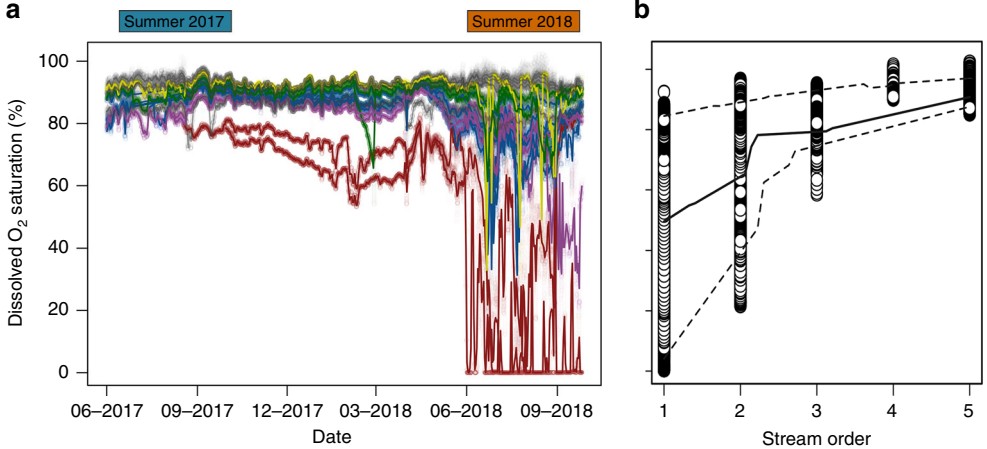

**Fig. 5 Network-scale drought altered the stream $O_2$ availability. a** Time series of stream-dissolved $O_2$ saturation (%) measured at 10-min intervals in the surface water of 16 stream sites across the KCS from June 2017 to October 2018 (shaded circles). Colored lines denote daily averages of $O_2$ saturation for different headwater streams (i.e., stream order 1 or 2; catchment area < 1.5 km$^2$; $n = 9$; Supplementary Fig. 1a, Table 1). Gray lines denote daily averages of dissolved $O_2$ saturation for higher-order streams (i.e., stream order > 2; catchment area > 1.5 km$^2$; $n = 7$; Supplementary Fig. 1a, Table 1). **b** Relationship between stream order and dissolved $O_2$ saturation for the same sites during summer 2018. Lines represent nonparametric 10th (lower dashed line), 50th (solid line), and 90th (higher dashed line) percentile regressions for the entire data set.

during this drought, particularly in low-order streams draining headwater catchments (Fig. 5). The magnitude and duration of episodic declines in stream $O_2$ concentrations varied across catchments, but all nine headwater streams had at least one documented excursion of $O_2$ below 50% saturation, five streams had one or more excursions below 25% saturation, and one remained anoxic for a large part of the summer. Hypoxic events, as well as their consequences for aquatic life, are well documented for estuaries and oceans[43], lakes[44], and large rivers[45,46], but in contrast, have been less of a focus in small streams. However, our results highlight drought as a mechanism that has the potential to cause $O_2$ stress in streams draining carbon-rich, headwater catchments. Importantly, these effects may be spatially widespread: for example, our results indicate potentially severe $O_2$ stress for all streams draining catchments smaller than 2 km$^2$ (Fig. 5b), which account for 65–80% of the total stream length in the KCS. Indeed, these small streams account for a large fraction of the drainage length in most biomes[47], yet are poorly represented by monitoring programs[48]. Our results suggest that capturing the spatial extent and significance of drought effects at high latitudes will require a shift in focus to these ecologically and biogeochemically vital environments.

The availability of $O_2$ affects biogeochemical processes and the associated cycling of macro- and micronutrients in aquatic ecosystems[43]. To better understand the consequences of deoxygenation for the metabolic balance of high-latitude streams, we evaluated temporal dynamics of the $CH_4$:$CO_2$ ratio in response to extreme low flows. By removing the influence of physical processes that affect the absolute concentration of both gases (e.g., reaeration), the ratio between $CH_4$ and $CO_2$ represents a useful proxy for methanogenesis in aquatic systems[35,49]. We assessed this ratio during 2017 and 2018 at five headwater streams (C1 –C7 in Supplementary Fig. 1a and Supplementary Table 1), and complemented this data set with similar observations made at 22 additional headwater locations during summer 2018 (Supplementary Fig. 1a). The results of this analysis show that the seasonal and episodic reductions in stream $O_2$ during the 2018 drought corresponded to significant increases in $CH_4$:$CO_2$ ratios (median of 0.014) compared with the previous summer (median of 0.0054; Supplementary Fig. 7). Furthermore, analysis of historical data from the same streams confirmed that, although the 2018 drought

was especially severe, past transitions to low flows during summer also led to increases in the dispersion and magnitude of stream $CH_4$:$CO_2$ ratios (Fig. 6). These findings are consistent with elevated rates of methanogenesis during low-flow periods[35]. High $CH_4$:$CO_2$ ratios (>0.1) are often associated with streams receiving anthropogenic nutrient inputs, whereas lower values (<0.0001) are more common for relatively undisturbed boreal or temperate forests[35]. Our results show that anthropogenic enrichment is not required to elevate this ratio, and instead indicate that climate-mediated pressure (i.e., seasonal drought events) at high latitudes may routinely alter the metabolic character of headwater environments, favoring methane production.

Increasing concentrations of $CO_2$ and $CH_4$ in response to drought may also influence the role that high-latitude streams play as sources of GHG to the atmosphere. While the contribution of small streams to network or regional GHG budgets is well studied in high-latitude landscapes[50,51], the controls over C gas evasion during drought remain largely unexplored. C gas evasion is the product of the concentration gradient between the stream and the atmosphere and the gas transfer velocity, and these parameters likely respond differently to drought. To explore this interplay, we estimated daily $CO_2$ and $CH_4$ flux across the water–air interface during summer periods (Supplementary Methods 2), and evaluated how these are influenced by discharge variation across five headwater streams in the KCS (C1–C7 in Supplementary Fig. 1a and Supplementary Table 1). Despite predictable declines in reaeration during low-flow periods (measured as $k_{600}$; Supplementary Fig. 8a and Supplementary Table 4), only $CO_2$ fluxes varied modestly with discharge (Supplementary Fig. 8b, Supplementary Table 4), indicating that low reaeration rates during drought constrained evasion losses. By comparison, $CH_4$ fluxes remained stable across the full discharge range due to the elevated concentration gradient between the stream and the atmosphere at low flows (Supplementary Fig. 8c, Supplementary Table 4), suggesting that rates of in-stream $CH_4$ supply during drought were sufficiently high to overcome the effects of reduced turbulence. Based on these results, we suggest that $CH_4$, which has an ~30-fold higher global warming potential than $CO_2$[8], is an emergent component of GHG budgets for headwater streams during these low-flow periods. This situation could be magnified when such events are

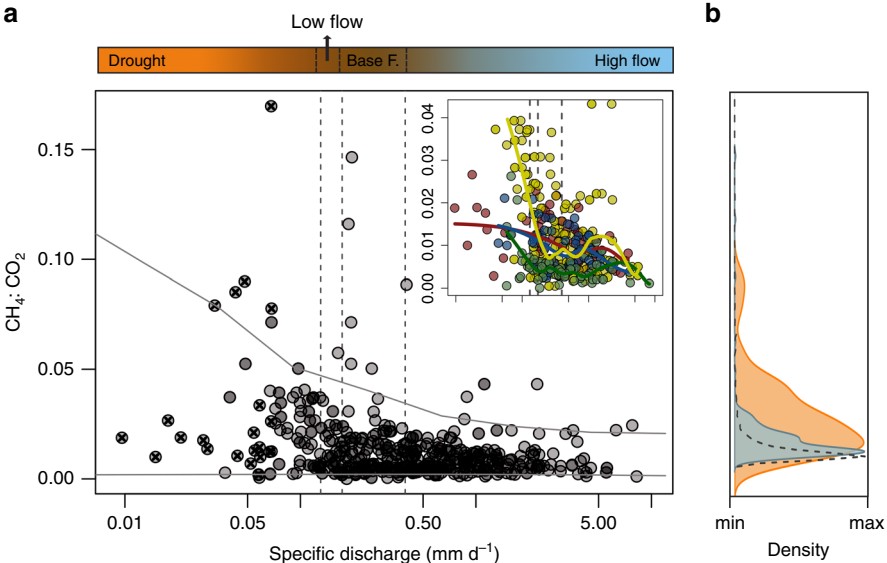

**Fig. 6 Drought promotes anaerobic signals in boreal headwater streams. a** Relationship between specific discharge (mm d$^{-1}$) and the molar ratio between $CH_4$ and $CO_2$ in the surface water of five headwater streams of the KCS (i.e., stream order 1 or 2; catchment area < 1.5 km$^2$; $n = 5$; Supplementary Fig. 1a, Table 1) during the summer period between January 2010 and October 2018. Observations with asterisks correspond to the 2018 drought. Vertical dashed bars represent thresholds among flow conditions during this period. Discharge delineation is based on percentile distributions of the historical (1980–2018) records in the catchment: drought (0th–10th percentile; $n = 59$), low flow (10th–20th percentile; $n = 22$), baseflow (20th–50th percentile; $n = 90$), and high flow (50th–100th percentile; $n = 193$). Gray lines are the nonparametric 10th and 90th percentile regression based on all of the data. The inset shows locally weighted regression model lines (Loess) for each headwater stream. **b** Density distributions for $CH_4:CO_2$ during drought (orange) and nondrought (blue) hydrological conditions. The dashed line represents the density distribution of stream $CH_4:CO_2$ from seven additional headwater catchments in northern Sweden sampled before the summer 2018 severe drought (site locations in Supplementary Fig. 1a)[50].

terminated by the resumption of flows that flush stream and near-stream environments. Further, these responses to drought in the headwaters stand in strong contrast to the increasing $CH_4$ sink strength in adjacent terrestrial habitats during such events, when low water tables limit production and facilitate $CH_4$ oxidation in soils[52–54]. Thus, $CH_4$ production and evasion from streams may account for a larger proportion of the catchment $CH_4$ production during years of severe drought.

**Drought in high-latitude landscapes**. Experimental and natural drought induced similar biogeochemical responses in KCS headwaters, yet differences in the severity and duration of effects among sites highlight how landscape context can mediate the propagation of this disturbance in boreal landscapes. First, at the smallest scales, variation in groundwater–stream connections governs local hydrological and chemical conditions as drought commences. Such effects were evident from the gradient in drought severity generated by our field experiment, but are also likely to be important under natural conditions, as variation in catchment topography and riparian soil volume determines the arrangement and persistence of hydrological connections over time[30]. At broader spatial scales, differences in land cover, topography, and soil characteristics influence patterns of runoff among headwaters by regulating the rates of evapotranspiration and water storage. For example, specific discharge during summer can differ by more than twofold among KCS streams, decreasing with tree volume, and increasing with soil depth and mire cover[55]. Such differences are most pronounced in dry years[55], and small streams draining till soils that support dense forests are likely the most vulnerable to extreme drought. Further, while mires may buffer drought effects on downstream waters, they are also strong sources of DOM and other reduced solutes and gases[56–58], and thus may promote drought-like chemical signals in streams even if hydrological conditions are less severe. Finally,

lakes are also abundant in northern landscapes and operate as important water-storage pools that, depending on their volume and arrangement, may alleviate or exacerbate drought effects downstream[59]. Overall, while current projections suggest that drought frequency may increase in northern Scandinavia[9] and parts of Canada[60], the consequences for streams will emerge from a complex and interacting set of biophysical factors that are likely to follow different trajectories across northern regions[4]. Predicting the future occurrence and severity of drought in these stream networks will require that we resolve how such interactions respond to ongoing climate change.

While we focused on the biogeochemical responses to drought, such events also have clear implications for aquatic communities and food webs in northern streams. It is evident that complete channel drying has catastrophic effects on aquatic communities[7]; however, the more widespread effect of drought observed here was the emergence of stagnant surface streams that remained hypoxic or anoxic for days or even weeks during summer 2018. Depending on their severity and duration, these low $O_2$ events can be lethal to many aquatic taxa, leading to reductions in overall biodiversity, and shifts in the composition of stream communities[61,62]. A host of unanswered questions remain regarding the impact of severe drought on high-latitude aquatic communities, including the mechanisms and timescales over which different taxonomic groups may recover. Suffice it to say, increases in the frequency of these events in northern landscapes could permanently reshape the biotic structure of headwaters.

In this era of climatic volatility, projected increases in the frequency and severity of droughts will alter the functional roles that streams and rivers play worldwide. Yet, little is known about the consequences of extreme low flows for stream networks draining high-latitude landscapes. This knowledge gap may reflect the less-frequent occurrence of drought at high latitudes historically, and/or the perception that such events are not severe enough to affect cold and humid regions. However, the responses

we document to the extreme 2018 drought in northern Europe challenge such assumptions. Together with the results from a manipulation experiment and historical stream chemistry data, these results suggest that drought in northern headwaters induces biogeochemical responses in streams that can trigger poor water-quality conditions across drainage systems. Increasing the occurrence of these events in northern regions would likely have major consequences for headwater streams, including the biogeochemical roles they play in landscapes and the ecosystem services they provide.

## Methods

**The reach-scale manipulation experiment.** The reach-scale hydrological manipulation was carried out during August 2017 in a 1.4-km headwater stream located at the upper section of the Krycklan Catchment Study (KCS)[28], in northern Sweden (Supplementary Fig. 1). The catchment draining the experimental reach is largely forested, with soils primarily composed of organic-rich deposits in low-lying areas and along the stream[29]. The experiment was divided in two periods: the drought period (from August 7th to 18th), achieved by damming an upstream lake (Supplementary Fig. 1b), and the background period, which comprised the period before (from August 3rd to 7th) and after (from August 24th to 30th) the drought manipulation (Supplementary Fig. 2). To capture the widest spectrum of responses along the reach, we selected six 50-m segments distributed along the stream (Supplementary Fig. 1b).

At the top and the bottom of the 1.4-km stream (Supplementary Fig. 1b), flumes are installed to estimate hourly discharge ($Q_{C5}$ and $Q_{C6}$, respectively; $m^3 s^{-1}$) based on 10-min water-level observations and stage-discharge rating curves developed from manual discharge measurements[55]. Hourly stream discharge was estimated every 50 m along the stream using a 2-m digital elevation model as $Q_i = (UCA_i/A_{C6-C5}) \times (Q_{C6}-Q_{C5})$, where $Q_i$ is stream discharge at channel grid cell $i$, $UCA_i$ is the upslope-contributing area along the stream channel at cell $i$ ($m^2$), and $A_{C6-C5}$ is the catchment area at C5 subtracted by the catchment area at C6 ($m^2$). The net groundwater inflow to each 50-m grid cell ($Q_{gw,i}$) was estimated as $Q_i - Q_{i-1}$. Previous studies using hydrologic tracers and hydrometric measurements suggest that this approach provides reasonable estimates of discharge and groundwater inflows along this study reach[25,30]. We assigned a discharge value for each study segment ($Q_S$; $m^3 s^{-1}$) from the modeled estimate. Likewise, we obtained lateral groundwater inputs entering into each 50-m segment ($G_S$; $m^3 s^{-1}$) from the difference between modeled discharge at the top and bottom of the segments.

Further, we obtained the mean stream depth ($z$; m) and wetted width ($w$; m) at the six segments (Supplementary Fig. 1b; Supplementary Table 2) from five cross-sectional transects along each segment (cross-sectional measurements every 10 cm). We then combined the segment-specific $z$ and $w$ with the $Q_S$ data to obtain the mean water velocity ($u = Q/z \times w$; $m s^{-1}$) for each segment. We derived the stream water residence time (WRT, min) for the six segments at hourly resolution by dividing the segment length (i.e., 50 m) by the mean water velocity. We chose WRT as the hydrological organizer because it correlates with a variety of functional metrics, such as DOM decomposition and chemistry[63,64], in-stream metabolism[65], hypoxia development[66], and nutrient uptake and delivery rates[67]. Note that estimates of Q and WRT were similar (± 10%; $n > 50$) to those obtained from salt releases made throughout the experiment period and previous studies[25] at different locations along the stream.

**Discrete sampling during the experiment.** We manually collected surface stream and hyporheic water at the bottom of each segment on five occasions (three during drought and two during background conditions, Supplementary Fig. 2b) to determine the concentrations of major electron acceptors (i.e., dissolved oxygen ($O_2$), nitrate ($NO_3^-$), sulfate ($SO_4^{2-}$), and carbon dioxide ($CO_2$)), major reduced products (i.e., ammonium ($NH_4^+$) and methane ($CH_4$)), dissolved organic carbon (DOC), as well as a set of basic physicochemical parameters (i.e, temperature, pH, and conductivity). Hyporheic samples were collected from 0.6- to 1.5-m-long PVC wells (10-cm Ø, screen length = 10–15 cm) installed in the hyporheic zone (depth = 25–50 cm) using a peristaltic pump. To minimize pumping effects and avoid artificial gas exchange, we pumped slowly and limited our withdrawal of water to a maximum of 250 ml per well. We additionally installed near-stream groundwater wells (depth = 50–100 cm) at the four main groundwater input zones discharging into the stream[25] and sampled them using the same methodology described for the hyporheic wells. For each water sample, we measured in situ conductivity, temperature, and $O_2$ concentration with portable meters (YSI, CA, USA). Samples for pH were collected in high-density polyethylene bottles, and filled completely without air bubbles. For DOC, $SO_4^{2-}$, $NO_3^-$, and $NH_4^+$ analysis, samples were filtered (0.45 μm) in the field and collected into clean, pre-rinsed polyethylene bottles. For $CO_2$ and $CH_4$, a separate 5-ml sample of bubble-free water was taken and injected into a 22.5-ml glass vial (containing nitrogen gas at atmospheric pressure) sealed with a rubber septum. The vials were prefilled with 0.5 ml of 0.6% HCl to shift the carbonate equilibrium toward $CO_2$. Samples were kept cold (for pH, DOC, $CO_2$, and $CH_4$) or frozen (for $NO_3^-$, $NH_4^+$, and $SO_4^{2-}$) until laboratory analyses.

**High-frequency sampling during the experiment.** At the bottom of each study segment, we measured continuous surface and hyporheic water $O_2$ concentration ($mg L^{-1}$), $O_2$ saturation (%), and temperature (°C) at 10-min intervals during the course of the experiment using MiniDOT loggers (PME, USA). In addition, at four of these six segments (S3–S6, Supplementary Fig. 1b), we also measured dissolved concentrations of $CO_2$ at the same frequency with a Vaisala GMT220 sensor (Vaisala, Finland) covered with a highly permeable membrane to dissolved gases but not to water[68] and connected to CR1000 data loggers (Campbell Scientific, Canada). Hyporheic sensors were placed in the same wells where the low-frequency sampling of stream hyporheic water was performed (see previous section). We used the continuous $O_2$ data to both validate low-frequency discrete $O_2$ observations and to model stream metabolism (see below and in Supplementary Methods 1).

**Network-scale monitoring.** To cover the widest spectrum of drainage sizes in the KCS network, we addressed chemical patterns in ten streams ranging from channel order 1 to 5 (sub-catchment drainage area from 0.04 to 68.9 $km^2$; circles with numbers in Supplementary Fig. 1a; Supplementary Table 1). Similarly, to cover a wide range of environmental conditions, we selected streams that drain distinct land covers that are representative of northern boreal landscapes, including forests, mires, and lakes (Supplementary Table 1). For the analyses, we grouped these ten streams into those draining headwater catchments (i.e., stream order 1 or 2; catchment area <1.5 $km^2$; $n = 5$; Supplementary Table 1) and those that do not (i.e., stream order > 2; catchment area > 1.5 $km^2$; $n = 5$; Supplementary Table 1). Thus, we used Strahler stream order to categorize these sites. Strahler stream order correlates with a variety of geomorphological metrics, including catchment drainage area[69] or stream width[70], and is thus a useful organizer for assessing patterns at the network scale[71].

At the ten monitoring stations and during two consecutive summers (2017 and 2018), we measured surface water $O_2$ concentration, $O_2$ saturation, and temperature at 10-min intervals with mniDOT loggers (PME, USA), and manually sampled for $CO_2$, $CH_4$, and DOC monthly (during winter) and every week (during summer and fall). In total, low-frequency chemistry data used for the analysis of the period between 2017 and 2018 derived from ~30 sampling occasions at each stream. In addition, to increase the spatial coverage of high-frequency $O_2$ data, we also deployed $O_2$ sensors in six different locations (four headwater catchments and two larger catchments; circles without number in Supplementary Fig. 1) during the two consecutive summers (2017 and 2018). Similarly, to increase the spatial resolution of low-frequency chemistry data (i.e., $O_2$, $CO_2$, and $CH_4$), we also carried out three synoptic surveys at 22 headwater streams of the KCS during the summer 2018 severe natural drought (triangles in Supplementary Fig. 1a).

**Long-term monitoring.** Apart from the 2017–2018 network monitoring, we also compiled monitoring data for $CO_2$ and $CH_4$ with a suite of additional chemical and physical parameters for ~9 consecutive years (2010–2018) at the same ten stream-monitoring stations in the KCS (square symbols in Supplementary Fig. 1a; Supplementary Table 1). Long-term monitoring samples were collected monthly during winter and every second week during summer and fall. This time period includes two consecutive summers (2017 and 2018) when sensor $O_2$ data were also recorded. We separated the summer period from the bulk long-term series based on historical (1980–2008) seasonal records in the catchment[42]. In total, noncontinuous chemistry data used for the analysis of the period between 2010 and 2018 derived from ~100 sampling occasions at each stream. In addition, at each of the ten monitoring stations, we measured Q at hourly intervals using a permanent H-flume[55]. To normalize and compare Q from the studied streams with different catchment areas, we report specific discharge (mm $day^{-1}$). To isolate drought hydrological conditions from the rest of the periods, we delineated the specific discharge and grouped studied responses based on percentile distributions of the historical (1980–2018) discharge records in the catchment[32]: drought (0th–10th percentile; $n = 59$), low flow (10th–20th percentile; $n = 22$), baseflow (20th–50th percentile; $n = 90$), and high flow (50th–100th percentile; $n = 193$).

**Laboratory analysis.** pH was measured using an Orion 9272 pH meter equipped with a Ross 8102 low-conductivity combination electrode with gentle stirring at ambient temperature (20 °C). DOC was analyzed by combustion using a Shimadzu TOC-V$_{PCH}$ (Shimadzu, Kyoto, Japan) following acidification to remove inorganic carbon. $NH_4^+$ and $NO_3^-$ were analyzed following the methods G-171-96 Rev.12 and Method G-384-08 Rev.2, respectively, with a SEAL Analytical AutoAnalyzer 3 (SEAL Analytical, WI, USA). $SO_4^{2-}$ was analyzed by liquid chromatography using a Metrohm IC Net 2.3 (Herisau, Switzerland). Finally, the concentration of $CO_2$ and $CH_4$ in the headspace gas samples was determined using a GC-FID Perkin-Elmer Clarus 500 (Waltham, MA, USA) equipped with a methanizer operating at 250 °C and connected to an autosampler Perkin-Elmer Turbo Matrix 110 (Waltham, MA, USA). Concentrations of other species of the DIC system (i.e., $HCO_3^-$ and $CO_3^{2-}$) were also determined using the stream pH, equations for carbonate equilibrium, and Henry's Law[72]. Free dissolved $CO_2$ was the predominant DIC form, accounting for >95% of DIC. Accordingly, $HCO_3^-$ and $CO_3^{2-}$ were discarded from the analysis due to their minor contribution to the overall DIC composition.

**Data treatment and statistical analyses.** For each study segment and experiment day, we used continuous $O_2$ measurements to estimate gross primary production (GPP) and ecosystem respiration (ER) with the open-channel single-station method[73]. We used Bayesian inverse modeling to estimate both GPP and ER[74,75]. A more detailed description of the stream metabolism modeling, quality assessment, and potential uncertainties can be found in Supplementary Methods. Note that here we only focused on ER, which is an integrative estimate of the ecosystem aerobic respiration occurring in the stream. We compared ER rates between drought and background periods using a nonparametric Wilcoxon Signed-Rank test. Further, we evaluated the relationship between ER and stream WRT to test the effects of drought on in-stream aerobic respiration.

To explore whether drought influenced the overall distribution of redox-sensitive solutes and gases in the stream, we built a principal component analysis (PCA) with the surface and hyporheic water $O_2$, $CH_4$, $SO_4^{2-}$, $NO_3^-$, and $NH_4^+$ concentrations from the samples collected during the experiment. We evaluated the dependency of the resulting scores of the PC1 (dependant variable) on stream WRT (independent variable) using linear and nonlinear regression models. We selected and reported the model with a higher coefficient of determination ($r^2$). Differences in the distribution of surface water PC1 scores between background and drought conditions were visually inspected with Kernel density plots and statistically tested using the nonparametric Wilcoxon Signed-Rank test.

We additionally assessed specific molar ratios of reduced to oxidized chemical forms (i.e., $NH_4^+$:$NO_3^-$ and $CH_4$:$O_2$). The $NH_4^+$:$NO_3^-$ ratio provides insight into the potential redox-driven changes in nitrogen cycling. Accordingly, an accumulation of nitrogen as $NH_4^+$ rather than $NO_3^-$ represents constraints on nitrification and increasing rates of denitrification[34]. Demand for $NO_3^-$ under reducing conditions is very high, as $NO_3^-$ is the most energetically favorable electron acceptor in the absence of oxygen[34]. We also used the $CH_4$:$O_2$ ratio to provide a synthesis of the full range of terminal electron-accepting processes in the sample. Low values of the $CH_4$:$O_2$ ratio indicate that aerobic pathways dominate the metabolic balance, while increases of $CH_4$:$O_2$ ratios represent a shift toward a dominance of anaerobic over aerobic metabolic processes[36,37]. To examine whether drought drove similar redox responses under experimental and natural conditions, we compared the relationship between stream WRT and surface water $CH_4$:$O_2$ molar ratios along the experimental stream reach during summers 2017 (experimental drought) and 2018 (severe natural drought).

To explore the influence of drought on the stream metabolic balance during the experiment, we compared molar deviations of $O_2$ and $CO_2$ from atmospheric equilibrium ($\Delta O_2$ and $\Delta CO_2$, respectively) for the discrete and high-frequency observations. The stoichiometry between $O_2$ and $CO_2$, in aquatic ecosystems is of particular interest because it provides insight into the dominance of the different metabolic pathways involved in the production and consumption of organic matter[76]. For instance, aerobic respiration of organic matter normally leads to $\Delta O_2$ and $\Delta CO_2$ relationships falling around the 1:–1 line. Deviations from this stoichiometry can be attributed to nonbiological processes (i.e., interactions of $CO_2$ with the carbonate system[77]) or to anaerobic respiratory pathways that produce $CO_2$ and $CH_4$ without consuming $O_2$[78]. We calculated $\Delta O_2$ and $\Delta CO_2$ from differences between the measured aqueous concentration of the gas ($C_w$) and its concentration in equilibrium with the atmosphere ($C_a$). Equilibrium concentrations were calculated from temperature and barometric pressure[72]. Different statistical analyses on $\Delta CO_2$:$\Delta O_2$ observations were used to test the treatment effects on the central tendency and dispersion of these data[36].

To assess the network-scale effects of the 2018 severe drought on the stream surface $O_2$ availability, we compared the 10-min $O_2$ saturation dynamics at the 16 stream-monitoring stations during 2017 and 2018. We evaluate the effect of catchment size (as stream order) on stream surface $O_2$ availability using nonparametric 10th, 50th, and 90th percentile regression. For this, each relationship was computed and plotted as a representation of the central tendency and dispersion of all the data. In addition, we explored patterns for the molar $CH_4$:$CO_2$ ratio during 2017 and 2018 to quantify the extent to which drought induced methanogenesis. Given that the quantification of whole-stream anaerobic metabolism is more difficult than aerobic respiration as a routine part of metabolism studies[35], molar ratios between $CH_4$ and $CO_2$ have been proposed as an indicator of methanogenesis in aquatic ecosystems[35,49]. Although this approach only provides a proxy of process rates, the use of ratios instead of absolute concentrations allows us to isolate the effect of physical processes affecting the absolute concentration of gases (e.g., hydrological mixing or atmospheric reaeration). Finally, because $CO_2$ and $CH_4$ measurements are currently more common than $O_2$ measurements in Swedish monitoring programs, using this ratio allowed us to more broadly assess the influence of drought on stream biogeochemistry.

To explore whether the summer 2018 stream biogeochemical responses extended to past low- flow periods, we analyzed the relationship between specific discharge and the molar $CH_4$:$CO_2$ ratio at the surface water of five headwater streams draining contrasted boreal catchments during summer for the period compressed between January 2010 and October 2018. Nonparametric 10th, 50th, and 90th percentile regression for all sites was computed and plotted as a representation of the central tendency and dispersion of the data. Specific site-to-site responses to different discharge levels were assessed with the locally weighted regression model (Loess). Differences in the distribution of surface water $CH_4$:$CO_2$ ratio across contrasting discharge conditions were visually inspected with Kernel density plots and statistically tested using the nonparametric Wilcoxon Signed-Rank test. Finally, to examine the significance of drought in promoting anaerobic respiratory processes at a wider regional level, we compared the $CH_4$:$CO_2$ ratio in the KCS during drought with the $CH_4$:$CO_2$ ratio of seven additional headwater catchments in Northern Sweden sampled before the summer 2018 severe drought and used as a reference for nondrought conditions (Supplementary Fig. 1a).

All statistical analyses were conducted with the R statistical environment (R Core Team 2018), except for PCA analysis, which was done with the software XLSTAT (XLSTAT 2019.1, Addinsoft SRAL, Germany). In R, we used the packages "stats", "nlme", and "vegan" to calculate and visualize linear and nonlinear regression models as well as nonparametric Wilcoxon Signed-Rank tests. We also used the "quantreg.nonpar" package to compute and visualize nonparametric percentile regressions. Statistical tests were considered significant when $p < 0.05$.

## Data availability

Meteorological data and maps for the summer 2018 drought are available in the Copernicus Climate Change Service (https://climate.copernicus.eu/) and the Swedish Meteorological and Hydrological Institute (https://www.smhi.se/en) portals, respectively. The data sets (Datasets_Gómez-Gener et al., 2020_NCOMMS; https://doi.org/10.6084/m9.figshare.11448513) have been deposited in Figshare Digital Repository https://figshare.com.

## Code availability

The R code used to generate the results (R_Scripts_Gómez-Gener et al., 2020_NCOMMS; https://doi.org/10.6084/m9.figshare.11448480), including step-by-step explanations of the statistical tests, has been deposited in Figshare Digital Repository https://figshare.com.

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

## Acknowledgements

This study was funded by the Swedish Research Foundation (VR) through SITES, Future Forests, Kempe Foundation, FOMA (SLU), Formas, SKB, and KAW. A.L. was supported by a Kempe Foundation stipend and a Juan de la Cierva grant (FJCI-2016-28416). The authors are thankful to Peder Blomkvist, Kim Lindgren, Abdulmajid Mahomoud, Ida Taberman, Johannes Tiwari, Åsa Boily and Stefan Ploum, and Nicolai Brekenfeld and Stefan Krause for field, lab, and database assistance. Jason Leach provided the groundwater data used in the paper. George Allen calculated the surface area and relative proportion of river and stream network above 50°N latitude.

## Author contributions

L.G., A.L., H.L., and R.S. designed the field experiment. L.G. and A.L. did the field work. L.G. performed the data analysis and generated all figures. L.G. and R.S. wrote the initial version of the paper with contributions from all the coauthors. All authors contributed to interpreting the results, discussing the associated dynamics, and developing the paper.

## Competing interests

The authors declare no competing interests.
