## [Peer Review File · Nature Communications]

Reviewers' comments:

Reviewer #1 (Remarks to the Author):

The manuscript reports results of (a) a short term manipulation of a small stream system to induce a hydrological drought and (b) an opportunistic sampling of a natural drought in 2018. The aims of the experiment are valid and worthwhile however I felt the data analysis and interpretation was not of sufficient quality. Specifically, I found the assertions made about anaerobic vs aerobic dominance changes in drought based on CO₂ and CH₄ concentration ratio changes problematic. Dissolved oxygen is present in the system which does not support assumptions of a switch to system-wide anaerobic dominance and the timescales involved (e.g. average increase of residence time from 38 to 264 min in short term experiment) are very short in relation to typical timeframes of microbial reduction sequences. The potential role of reduced dilution of groundwater fluxes as a mechanism driving change is not explored adequately. Perhaps as a consequence the Principle Component Analysis and some of the other graphs are not convincing about the stated differences between drought and background conditions. Further specific comments are provided below.

Line 34-35 – On hydrological droughts – I feel it would also be worthwhile to mention that hydrological droughts can be caused by human activities independent or semi-independent to climate (e.g. via water diversions or damming of river systems).

Figure 1 – why is the interquartile range changing with time – I thought this should be horizontal lines for this range across whole dataset?

Figure 2 a. I found the PCA presented in Figure 2 not particularly convincing. The drought and background samples cluster together for many of the components, with the possible exception of NH₄ for 3 samples. The statement about reduced vs oxidised state on this Figure is also misleading in the absence of redox potential and pH data.

Line 89 Statement that “Furthermore, measurements from the same locations along this stream reach during the extreme natural drought of 2018 revealed nearly identical relationships between stream chemistry and WRT (Fig. S3).” I disagree with this statement, there is a two order of magnitude difference in ratios at low water residence time.

Line 113 – Statement that “Indeed, analysis CH₄:CO₂ ratios during the experiment (Fig. S6) indicate that, where reductions in flow were most severe, methanogenesis, the least energetically favorable microbial processes dominated over aerobic metabolism” The surface water shows very little change which does not support this statement. If DO was present as measured, how could methanogenesis occur in surface water?

Figure 3 Statement that “drought promoted higher and more dispersed $\Delta\text{CO}_2:\Delta\text{O}_2$ observation clouds compared to background conditions (Fig. 3), indicating that anaerobic processing of DOM, which generates excess CO₂ (Fig. 3a) and CH₄ (Fig. 3b) relative to O₂, dominated the metabolic balance during drought.” I am not completely sure what an “observation cloud” is but I think this refers to the scatter of the data and the authors have drawn a “cloud” around it. In any case there is substantial overlap between drought and non-drought again which leads to some doubts around the stated cause (anaerobic DOM processing) and effects (excess CO₂ and CH₄) statements made. Also the DO data Figure S5 does not support the statement about anaerobic conditions as mentioned above. The residence time changes involved are relatively short compared with typical times to transition through reduction sequences, particularly to reach methanogenesis.

Hence it may be that groundwater input is less diluted during drought which gives an apparent effect in this regard? I see the authors have done some groundwater measurements, was there CH₄ present in groundwater and how could CH₄ concentrations in hyporrheic zone and surface water change during

drought conditions due to e.g. less groundwater recharge or less dilution in surface water.

Line 125 Statement that "Patterns of hypoxia, as well as its consequences for aquatic life, are well documented for lakes, estuaries and oceans, but are less of a focus in running waters" is incorrect. For example see numerous papers of Baldwin and colleagues for hypoxia in the Murray-Darling river system. The classic research on hypoxia by Streeter and Phelps in the 1930s was developed in running waters.

Figure 4 – why is there such variability in the sensor dataset in 2018, are the authors sure the sensors did not dry out during this time?

A way forward – I found the most convincing results were the stream metabolism data presented in Supplementary Material Figure S4. These show a consistent pattern with residence time and sharp contrast between drought and reference conditions. I would suggest a refocussing of paper to concentrate more on aerobic metabolism changes rather than trying to assert switches to anaerobic mechanisms which are difficult to prove based on the evidence provided (just CO₂:CH₄ gas concentration ratio shifts and maintenance of oxic-hypoxic conditions).

Reviewer #2 (Remarks to the Author):

Review of Drought-induced biogeochemical shifts in high latitude stream networks

By Emily S. Bernhardt, Duke University

In this MS, Gomez-Gener et al. report substantial shifts in the chemistry, energetics and biogeochemical cycling of surface and hyporheic waters of boreal forested streams as a result of the 2018 northern European drought. Subsequent experimental manipulation of streamflows and water residence time provides strong empirical support for the hypotheses proposed to explain the field patterns observed during the natural drought. This study is among the first to report the consequences of hydrologic drought for high latitude rivers, the results are important and the data and inferences are robust and sound. I would like to see this work published and believe it will be an important contribution to global change biology, ecosystem science and aquatic ecology.

That said, I found the structure of the paper a challenge. While the graphical data presentation is fantastic, the ordering of points and the description of results in the text forces the reader to do too much work and is likely to reduce the impact of the paper. I would like to encourage the authors to consider the following suggestions for increasing the impact of this impressive research.

1) Provide greater clarity about the ecosystem under study and its boundaries.

It is fortunate that I have been to the Krycklan catchments and have some idea of what these systems are like, but this MS does not describe the system under study. It never even mentions what the vegetation is like or anything about the slope or the abundance of wetlands. This context is critically important to the interpretation of your results. Do you believe your findings can apply to all "northern latitude" ecosystems or to all boreal forested wetlands? At present the only description of the study site is that it is Northern Sweden.

It is equally important to be very clear at the outset that this study is about the aquatic ecosystem and not the catchment. Your study ecosystem is a river network and this should be explained. All data come from streamwater and hyporheic samples. Again, the lack of a site description led me to first expect a catchment study and to be initially very surprised by the reduction in aerobic respiration from

a wetland dominated catchment upon drainage. You might want to point out this literature - its quite well established that drainage of wetlands and wet soils can enhance aerobic respiration and soil carbon loss. Here you are showing that the aquatic ecosystems draining those catchments may undergo quite different and even opposite trajectories of change.

Providing much greater clarity about the ecosystem to which these questions pertain and to which these findings might be scaled would substantially improve the impact of this paper.

2) Organize the findings in order of their complexity.

I found the presentation of results compelling but disordered. I would suggest that a stronger paper would build from the very obvious to the very exciting.

I would recommend the following flow

Drought reduces flow and increases water residence times - please say something about flows here in addition to WRTs - in general it would be useful for the authors to think about fluxes as well as pool sizes.

As WRTs increase, dissolved oxygen concentrations decline - many stagnant reaches become hypoxic or even anoxic

When we calculate aerobic metabolism from the diel O₂ signals we see that rates decline substantially during droughts.

Alongside this reduction in O₂ and aerobic metabolism, we see increased concentrations of CH₄, a shift to high NH₄:NO₃ ratios and increases in both the CO₂:O₂ and CH₄ to O₂ ratios.

In the most severe cases, stream reaches shift to methanogenesis as the dominant metabolic pathway. Explain more carefully the work done to document that methanogenesis becomes a greater energy source than aerobic respiration... I thought this was THE most exciting finding but the way you determined this from the CH₄:CO₂ ratios was not explained well enough to allow replication or synthesis

3) More carefully consider whether the higher concentrations of reduced gases indicate an increase in production or a reduction in export... You can get your increase in concentrations via either mechanism and whether total export is changing should matter quite a bit to your interpretations. If the former - there are landscape/regional implications, if the latter, this simply shifts the local environment but has little impact at larger scales. To rephrase this point, are you observing a catchment scale impact or an aquatic ecosystem only impact? In either case, its worth considering how and whether these in-stream impacts of drought interact with the likely impacts of drought in the uplands. I have a couple of thoughts in this regard

Does the composition of DOC inputs to these streams shift as a result of drought? I would expect the declining water tables would lead to more complete degradation of SOM and, perhaps, a reduction in complex DOC molecule export to streams... but perhaps the terrestrial and wetland components of these watersheds simply become disconnected by drought.

What are the net GHG consequences of the decline in aerobic respiration and increase in anaerobic metabolism? Does drought increase or reduce freshwater GHG emissions when you calculate them in CO₂ equivalents. I think this is an important question for making this study of broader interest to ecosystem ecologists.

What if anything is known about how the catchment C balance responds to drought? How does this new information contribute to or change that understanding (either in terms of the spatial heterogeneity of responses OR in terms of the total magnitude of ecosystem change).

Minor Comment

Lines 75-77 - I think there is a mistake here... did the WRT increase (as the #s suggest) or decrease (as the wording suggests). Spend a bit more time on this important point (since it is key to several of your graphs).

Reviewer #3 (Remarks to the Author):

This paper entitled "Drought-induced biogeochemical shifts in high latitude stream networks" by Gomez-Gener and others reports the effects of a manipulative drought experiment and a naturally dry year on aerobic and anaerobic biogeochemical processes. In both the manipulation and dry year, they found that in drought conditions, reduced compounds, including methane, were more abundant, correlated with water residence time. The study is unique in its spatial extent, demonstrating a network-wide response to drought that will be of broad interest to readers. Additionally, the coupling of a manipulation with a medium-term observational study gives added credibility to the conclusions. I have two minor concerns and several line edits, but after revision, I believe this study would be a valuable contribution to this journal.

1. My main criticism is that the authors attribute all the changes to water residence time without exploring alternative hypotheses for the increase in reduced compounds. How much of the drought effects are due directly to hydrological changes versus other, indirect factors? For example, reaeration of oxygen varies with flow (more turbulence), and temperature is typically warmer during low flow conditions (Brown & Hannah 2007; Klaus et al. 2019). Because oxygen affect aerobic respiration and temperature regulates all the cited processes, drought it would substantially alter not only the hydrological residence time (which is the main dynamic described by the authors), it would alter the biogeochemical rates and the exposure time to conditions conducive for the different reactions (Oldham et al. 2013; Kolbe et al. 2019). I assume the manipulative experiment would have less variation in temperature than the drought year, but residence time is still tightly coupled to water flowpath and biogeochemical conditions (Abbott et al. 2016).

2. This kind of distributed, watershed-scale approach can reveal the extent of change, but can also be leveraged to identify the drivers of change (two recent examples from the Arctic: Connolly et al. 2018; Shogren et al. 2019). Besides looking at the distribution of oxygen concentrations with stream size, I wonder if the authors could explore how different catchment characteristics increase or decrease the likelihood of shifts into anoxia with drought. Said otherwise, could the authors use the rich spatiotemporal data from the Krycklan study area to explore why some rivers experienced persistent anoxia, while others were largely unaffected (e.g. explain the variability observed in Fig. 4)?

Line edits:

Line 16: Because neither the experiment nor drought have been previously introduced, it is hard to parse this list.

44-59: This seems too detailed for this portion of the manuscript. Could a more general treatment of the relationship between residence time and biogeochemical conditions suffice here?

62: "Whether" and "if" are yes or no questions. "How much" and "to what degree" are richer and more informative.

75: Typo (increased rather than reduced). Given the subject of the paper, perhaps avoid reduced altogether, except when referring to chemistry (decreased is clearer in this context).

77: How comparable is level of drought severity with amount of flow in the stream? The terrestrial environment, which sets the template for the water chemistry and which becomes more important if upstream flow is cut off as was done here, is not drought stressed, presumably.

Lines 82 and onward: The level of detail in the results feels a little imbalanced. Some of them are very specific to this study (PCA axes), while others are more applicable across studies (residence time

versus redox state). Reorganizing and perhaps subdividing a little more could improve this.
119 and elsewhere: water chemistry is more specific than water quality
163-164: this seems like a valid hypothesis, but is it based on the results of the current study (as the sentence states) or the cited study?
166-167: Unclear. Potentially rephrase as "CH₄ production in aquatic environments could represent a larger proportion of ecosystem-level CH₄ balance during these years" or something like that.
168: typo "stimulates"
170: The primary product of denitrification, particularly in situations of long residence time, is the inert gas N₂. This can be an important ecosystem service in nutrient saturated environments, which could be emphasized as a positive tradeoff.
181-191: This seems largely redundant and could be removed without affecting the content of the paper.
Figure 4 is difficult to understand. Because it has different axes, the inset graph of the stream order might be more comprehensible as a separate panel.

Citations:

Abbott, B.W., Baranov, V., Mendoza-Lera, C., Nikolakopoulou, M., Harjung, A., Kolbe, T., et al. (2016). Using multi-tracer inference to move beyond single-catchment ecohydrology. *Earth-Sci. Rev.*, 160, 19–42.

Brown, L.E. & Hannah, D.M. (2007). Alpine Stream Temperature Response to Storm Events. *J. Hydrometeorol.*, 8, 952–967.

Connolly, C., Khosh, M.S., Burkart, G.A., Douglas, T.A., Holmes, R.M., Jacobson, A.D., et al. (2018). Watershed slope as a predictor of fluvial dissolved organic matter and nitrate concentrations across geographical space and catchment size in the Arctic. *Environ. Res. Lett.*

Klaus, M., Geibrink, E., Hotchkiss, E.R. & Karlsson, J. (2019). Listening to air–water gas exchange in running waters. *Limnol. Oceanogr. Methods*, 17, 395–414.

Kolbe, T., Dreuz, J.-R. de, Abbott, B.W., Aquilina, L., Babey, T., Green, C.T., et al. (2019). Stratification of reactivity determines nitrate removal in groundwater. *Proc. Natl. Acad. Sci.*, 201816892.

Oldham, C.E., Farrow, D.E. & Peiffer, S. (2013). A generalized Damköhler number for classifying material processing in hydrological systems. *Hydrol Earth Syst Sci*, 17, 1133–1148.

Shogren, A.J., Zarnetske, J.P., Abbott, B.W., Iannucci, F., Frei, R.J., Griffin, N.A., et al. (2019). Revealing biogeochemical signatures of Arctic landscapes with river chemistry. *Sci. Rep.*, 9, 1–11.

Reviewer #1 (Remarks to the Author):

The manuscript reports results of (a) a short term manipulation of a small stream system to induce a hydrological drought and (b) an opportunistic sampling of a natural drought in 2018. The aims of the experiment are valid and worthwhile however I felt the data analysis and interpretation was not of sufficient quality. Specifically, I found the assertions made about anaerobic vs aerobic dominance changes in drought based on CO₂ and CH₄ concentration ratio changes problematic. Dissolved oxygen is present in the system which does not support assumptions of a switch to system-wide anaerobic dominance and the timescales involved (e.g. average increase of residence time from 38 to 264 min in short term experiment) are very short in relation to typical timeframes of microbial reduction sequences. The potential role of reduced dilution of groundwater fluxes as a mechanism driving change is not explored adequately. Perhaps as a consequence the Principle Component Analysis and some of the other graphs are not convincing about the stated differences between drought and background conditions. Further specific comments are provided below.

Response: We appreciate that reviewer 1 considers this work worthwhile but also value the criticisms of the analysis and interpretation of results. Indeed, these comments prompted us to clarify several issues in the revised manuscript.

Line 34-35 – On hydrological droughts – I feel it would also be worthwhile to mention that hydrological droughts can be caused by human activities independent or semi-independent to climate (e.g. via water diversions or damming of river systems).

Response: We have followed this suggestion and added a sentence describing how droughts can be promoted anthropogenically (lines 32 to 34 of revised manuscript).

Figure 1 – why is the interquartile range changing with time – I thought this should be horizontal lines for this range across whole dataset?

Response: Here we use the interquartile range (IQR) of the discharge (i.e., the difference between 90th and 10th percentiles) as measure of statistical dispersion of the discharge for each day of the year (i.e., Julian day) for the period between 1981 and 2010. Accordingly, the IQR of the daily discharge is given by the inter-annual variability of this period for each specific day.

Figure 2 a. I found the PCA presented in Figure 2 not particularly convincing. The drought and background samples cluster together for many of the components, with the possible exception of NH₄⁺ for 3 samples. The statement about reduced vs oxidized state on this Figure is also misleading in the absence of redox potential and pH data.

Response: We appreciate the interpretation of this analysis provided by the reviewer.

The main goal of using PCA was to get a broad view of how experimental drought influenced the overall chemical patterns in our stream reach (including surface and

hyporheic environments). Compared to focus on individual solutes, this approach integrates variation among the electron acceptors and reduced products for which we have data. In the revised version of the PCA, which now includes pH as suggested by R1, the surface water observations between the two treatments do not cluster in almost any case (see Fig. S4 of revised manuscript). Consistent with this, the Wilcoxon Signed-Ranks test shows that surface water PC1 scores were significantly higher during drought. Regarding hyporheic sediments, while we acknowledge some overlap of background and drought samples, it is still apparent that drought promoted the accumulation of reduced relative to oxidized compounds in subsurface environments. For example, the PC1 scores for the hyporheic water was also significantly higher during drought than background conditions. Moreover, PC1 scores for surface and hyporheic water increased non-linearly among sites as drought intensity increased (measured as greater WRT).

Overall, the commentary made by the reviewer, as well as the results obtained from additional analysis stimulated by this comment, made us rethink how we present these results. In the revision, we have:

- 1) Removed the PCA figure from the main text and placed it in the SI (Fig. S4 of revised manuscript).
- 2) Only used the PCA results to infer on the distribution of redox-sensitive solutes and gases in the stream and not to directly interpret the relative dominance of anaerobic vs aerobic processes.
- 3) We have complemented the PCA results with independent analysis of molar $\text{NH}_4^+:\text{NO}_3^-$ and $\text{CH}_4:\text{O}_2$ ratios during the experiment (Fig. 3 of revised manuscript). We have chosen these ratios because they represent most contrasted chemical species in the PCA and provide insight into the potential redox-driven changes in both the nitrogen and carbon cycling.

Finally, we agree with the reviewer that the statement about reduced vs oxidized state might be misleading considering that the redox potential (E_h) was not measured in the experiment. Following his/her suggestion we have i) included pH in the PCA and ii) changed the statements “oxidized status” and “reduced status” by “oxidized solute forms” and “reduced solute forms” to avoid the use of a definition based on the redox state of the system but instead on the redox state of the studied solutes.

Line 89 Statement that “Furthermore, measurements from the same locations along this stream reach during the extreme natural drought of 2018 revealed nearly identical relationships between stream chemistry and WRT (Fig. S3).” I disagree with this statement, there is a two order of magnitude difference in ratios at low water residence time.

Response: This is also a really helpful comment. Although the comparison of $\text{CH}_4:\text{O}_2$ from the same locations between experimental (2017) and natural (2018) drought shows qualitatively similar relationships with WRT (i.e., the shapes of the curves are

similar), the reviewer correctly notes that the y axis values are quite different at low WRT. We attribute this difference to the effect of our drought manipulation on the contribution of lateral groundwater with high CH₄:O₂ ratios during early stages of the experiment. Specifically, our manipulation interrupted the water coming in from the top of the study reach, but did not induce drought on land, and thus did not initially influence lateral groundwater inflows (i.e., between riparian zones and stream). Thus, these lateral connections sustained discharge as well as solute and gas supply during the first phase of the experiment. This contribution declined as the experiment progressed and WRT increased – to the point that eventually we could no longer draw water from near-stream sampling wells (i.e., they went dry). By contrast, during natural drought (2018), both the longitudinal (i.e., water from upstream) and lateral hydrological connections were weakened at roughly the same time and pace, leading reduced methane supply to the stream throughout our measurement period.

However, this ratio ultimately converged as WRT increased and lateral groundwater inputs declined, suggesting that these chemical signals can be sustained by processes occurring within the stream/hyporheic zone. At the same time, the differences between these two curves across the full range of WRTs does provide an example of how variation in local groundwater hydrology during the onset of drought can potentially exacerbate transitions toward reducing chemical conditions in the stream through hydrologic inputs. To better describe the patterns observed for the CH₄:O₂ ratios during both the manipulation (2017) and the natural drought (2018), we have now included additional text and two supplementary figures showing:

- 1) The groundwater inflow dynamics to the stream during the field experiment (Fig. S3 of revised manuscript).
- 2) The relationships between WRT and groundwater CH₄:O₂ ratios (Fig. 3 of revised manuscript)

Line 113 – Statement that “Indeed, analysis CH₄:CO₂ ratios during the experiment (Fig. S6) indicate that, where reductions in flow were most severe, methanogenesis, the least energetically favorable microbial processes dominated over aerobic metabolism” The surface water shows very little change which does not support this statement. If DO was present as measured, how could methanogenesis occur in surface water?

Response: We agree that although surface DO decreased below the 50% saturation level at higher WRT, the generally high values during the experiment did, most likely, limit the likelihood of methanogenesis occurring in the surface water. In fact, it was not our intent to suggest methanogenesis in the surface stream. However, the reduction in DO was more visible (and statistically significant) in the hyporheic zone, where beyond ca. 200 min of WRT, levels dropped and remained below the critical saturation level of 25%. These low values, coupled with observed increases in CH₄ and CO₂ in the hyporheic sediments and surface stream, support the idea that drought induced elevated rates of anaerobic metabolism. Indeed, other studies have documented methanogenesis in hyporheic sediments (e.g., Baker et al., 1999; Jones et al., 2008),

so this is not in itself surprising. In the revised version, we have thus clarified the major role of the hyporheic habitats on controlling the drought-induced biogeochemical changes reported in the manuscript (e.g., see lines 99 to 102; or lines 111 to 113 of revised manuscript).

Figure 3 Statement that “drought promoted higher and more dispersed $\Delta\text{CO}_2:\Delta\text{O}_2$ observation clouds compared to background conditions (Fig. 3), indicating that anaerobic processing of DOM, which generates excess CO_2 (Fig. 3a) and CH_4 (Fig. 3b) relative to O_2 , dominated the metabolic balance during drought.” I am not completely sure what an “observation cloud” is but I think this refers to the scatter of the data and the authors have drawn a “cloud” around it. In any case there is substantial overlap between drought and non-drought again which leads to some doubts around the stated cause (anaerobic DOM processing) and effects (excess CO_2 and CH_4) statements made. Also the DO data Figure S5 does not support the statement about anaerobic conditions as mentioned above. The residence time changes involved are relatively short compared with typical times to transition through reduction sequences, particularly to reach methanogenesis.

Response: To rigorously analyze and interpret results from this stoichiometric analysis (Fig. 4), we have now used three metrics that described the central tendency and the dispersion of $\Delta\text{CO}_2:\Delta\text{O}_2$. These three metrics are based on a new synthesis of this analytical approach (Vachon et al., 2019) and were computed for surface and hyporheic samples prior to and during the manipulation. We report these metrics, together with a description of how to compute and interpret each, in a new supplementary Table (Table S3 of revised manuscript). In addition, we have also replaced the original panel b ($\Delta\text{CH}_4:\Delta\text{O}_2$), which we felt was too redundant to the previous results, with a panel showing high-frequency ΔCO_2 and ΔO_2 relationships based on surface stream data for one of the study segments (Fig. 4b of revised manuscript). This figure provides further evidence, at a far higher temporal resolution that drought promoted higher $\Delta\text{CO}_2:\Delta\text{O}_2$ as WRT increased throughout the experiment. The text has been expanded and updated accordingly (lines 135 to 148 of revised manuscript).

Finally, we are not completely sure how to interpret the question about whether residence times were sufficiently long for microbes to transition through terminal electron accepting processes. First, we do acknowledge that the early responses to the experiment (e.g., the first few days) were very likely influenced by groundwater inputs of reduced compounds (see comment above), but by the end the experiment (nearly 2 weeks), these responses seem to be no longer driven by hydrology. In addition, we use WRT time here to characterize the degree of low flow conditions among study reaches. We are not suggesting that the system transitioned through these different metabolic processes within (for example) 200 min. Instead, we argue that study reaches, which sustained really flow rates (described by the WRT) show clear chemical signals of low oxygen conditions and anaerobic process in near stream and hyporheic sediments.

Hence it may be that groundwater input is less diluted during drought which gives an apparent effect in this regard? I see the authors have done some groundwater measurements, was there CH₄ present in groundwater and how could CH₄ concentrations in hyporrheic zone and surface water change during drought conditions due to e.g. less groundwater recharge or less dilution in surface water.

Response: See comment above regarding groundwater dynamics during the experiment. Again, to clarify the potential effect of groundwater inputs during the experiment, we have included, described, and discussed both a figure with the groundwater hydrological dynamics during the experiment (Fig. S3 of revised manuscript) as well a figure with the relationships between WRT and stream, hyporheic and groundwater CH₄:O₂ ratios (Fig.3 of revised manuscript).

Line 125 Statement that “Patterns of hypoxia, as well as its consequences for aquatic life, are well documented for lakes, estuaries and oceans, but are less of a focus in running waters” is incorrect. For example see numerous papers of Baldwin and colleagues for hypoxia in the Murray-Darling river system. The classic research on hypoxia by Streeter and Phelps in the 1930s was developed in running waters.

Response: We have followed the suggestion made by the reviewer, modified the sentence accordingly and included an additional reference (lines 164 to 166 of revised manuscript).

Figure 4 – why is there such variability in the sensor dataset in 2018, are the authors sure the sensors did not dry out during this time?

Response: This variability is precisely a consequence of periods with very low flows that promoted episodic declines in stream O₂ concentrations during the 2018 summer drought. To better visualize the longitude and magnitude of low O₂ periods, daily average of dissolved oxygen saturation has additionally been incorporated to the new version of Fig.5.

Also, the high-frequency O₂ sensors used in the study (MiniDOT, PME, USA; <https://www.pme.com/products/minidot>) is an optode that measures dissolved oxygen concentration in water through a fluorescence method. Therefore, exposure of these sensors to the atmosphere (e.g., during episodes of stream drying) should give O₂ measurements around the atmospheric equilibrium instead of 0. In addition, we visited these sites frequently during the summer 2018 drought to ensure they remained inundated as the stream network contracted.

A way forward – I found the most convincing results were the stream metabolism data presented in Supplementary Material Figure S4. These show a consistent pattern with residence time and sharp contrast between drought and reference conditions. I would

suggest a refocussing of paper to concentrate more on aerobic metabolism changes rather than trying to assert switches to anaerobic mechanisms which are difficult to prove based on the evidence provided (just CO₂:CH₄ gas concentration ratio shifts and maintenance of oxic-hypoxic conditions).

Response: We have followed the suggestion made by the reviewer and included the aerobic metabolism data in the main manuscript (Fig. 2 of revised manuscript). Additionally, we have re-organized the results to give a greater weight to the drought-induced effects on the aerobic metabolism patterns. Again, we now interpret the drought-induced patterns as a diversification of metabolic processes or co-occurrence/co-existence of metabolic processes with different redox requirements rather than a change in the dominance. In the revised version, this has been changed consistently throughout the text.

References

- Baker, M. A., Dahm, C. N., & Valett, H. M. (1999). Acetate retention and metabolism in the hyporheic zone of a mountain stream. Limnology and Oceanography, 44(6), 1530–1539. <https://doi.org/10.4319/lo.1999.44.6.1530>*
- Jones, J. B., Holmes, R. M., Fisher, S. G., Grimm, N. B., & Greene, D. M. (2008). Methanogenesis in Arizona , USA dryland streams, 155–173.*
- Vachon, D., Sadro, S., Bogard, M. J., Lapierre, J. F., Baulch, H. M., Rusak, J. A., et al. (2019). Measuring coupled O₂ - CO₂ provide emergent insights into aquatic ecosystem function. Limnology and Oceanography Letters, In press.*

Reviewer #2 (Remarks to the Author):

By Emily S. Bernhardt, Duke University

In this MS, Gomez-Gener et al. report substantial shifts in the chemistry, energetics and biogeochemical cycling of surface and hyporheic waters of boreal forested streams as a result of the 2018 northern European drought. Subsequent experimental manipulation of streamflows and water residence time provides strong empirical support for the hypotheses proposed to explain the field patterns observed during the natural drought. This study is among the first to report the consequences of hydrologic drought for high latitude rivers, the results are important and the data and inferences are robust and sound. I would like to see this work published and believe it will be an important contribution to global change biology, ecosystem science and aquatic ecology.

Response: We appreciate the overall positive impression of our study.

That said, I found the structure of the paper a challenge. While the graphical data presentation is fantastic, the ordering of points and the description of results in the text forces the reader to do too much work and is likely to reduce the impact of the paper. I would like to encourage the authors to consider the following suggestions for increasing the impact of this impressive research.

Response: We thank the reviewer for providing such thoughtful and constructive suggestions for how to organize the manuscript to improve its impact. We have restructured the manuscript based on these suggestions (see below for specific responses for the different comments and suggestions raised by reviewer 2).

1) Provide greater clarity about the ecosystem under study and its boundaries:

It is fortunate that I have been to the Krycklan catchments and have some idea of what these systems are like, but this MS does not describe the system under study. It never even mentions what the vegetation is like or anything about the slope or the abundance of wetlands. This context is critically important to the interpretation of your results. Do you believe your findings can apply to all “northern latitude” ecosystems or to all boreal forested wetlands? At present the only description of the study site is that it is Northern Sweden. It is equally important to be very clear at the outset that this study is about the aquatic ecosystem and not the catchment. Your study ecosystem is a river network and this should be explained. All data come from streamwater and hyporheic samples. Again, the lack of a site description led me to first expect a catchment study and to be initially very surprised by the reduction in aerobic respiration from a wetland dominated catchment upon drainage. You might want to point out this literature - its quite well established that drainage of wetlands and wet soils can enhance aerobic respiration and soil carbon loss. Here you are showing that the aquatic ecosystems draining those catchments may undergo quite different and even opposite trajectories of change.

Providing much greater clarity about the ecosystem to which these questions pertain and to which these findings might be scaled would substantially improve the impact of this paper.

Response: To better describe the context and framing of the study system, we have included several new lines in the main text of the revised manuscript. Here we describe more extensively the location and typologies of the streams that we studied, as well as the vegetation and main landscape elements of the adjacent terrestrial and wetland ecosystems (lines 70 to 83 of revised manuscript). Additionally, we have also extended the description of the experimental stream reach (lines 83 to 85 of revised manuscript). Note that Table S1 and S2 has been included as supplementary information and contain additional quantitative information about the catchment and stream characteristics, respectively.

2) Organize the findings in order of their complexity:

I found the presentation of results compelling but disordered. I would suggest that a stronger paper would build from the very obvious to the very exciting.

I would recommend the following flow:

- Drought reduces flow and increases water residence times - please say something about flows here in addition to WRTs - in general it would be useful for the authors to think about fluxes as well as pool sizes.
- As WRTs increase, dissolved oxygen concentrations decline - many stagnant reaches become hypoxic or even anoxic
- When we calculate aerobic metabolism from the diel O₂ signals we see that rates decline substantially during droughts.
- Alongside this reduction in O₂ and aerobic metabolism, we see increased concentrations of CH₄, a shift to high NH₄:NO₃ ratios and increases in both the CO₂:O₂ and CH₄ to O₂ ratios.

In the most severe cases, stream reaches shift to methanogenesis as the dominant metabolic pathway. Explain more carefully the work done to document that methanogenesis becomes a greater energy source than aerobic respiration. I thought this was THE most exciting finding but the way you determined this from the CH₄:CO₂ ratios was not explained well enough to allow replication or synthesis.

Response: We thank again the reviewer for suggesting this structure. We have re-organized both the text and figures of the manuscript accordingly.

3) More carefully consider whether the higher concentrations of reduced gases indicate an increase in production or a reduction in export. You can get your increase in concentrations via either mechanism and whether total export is changing should matter quite a bit to your interpretations. If the former - there are landscape/regional implications, if the latter, this simply shifts the local environment but has little impact at

larger scales. To rephrase this point, are you observing a catchment scale impact or an aquatic ecosystem only impact? In either case, it's worth considering how and whether these in-stream impacts of drought interact with the likely impacts of drought in the uplands. I have a couple of thoughts in this regard.

Response: We agree that this is a critical point that needed more consideration. Thus, we have expanded the discussion about how the use of molar concentration ratios serve as a proxy for methanogenesis by isolating the effects of export (which affect CO₂ and CH₄ similarly; lines 175 and 179 of revised manuscript as well as Method section). Additionally, to explore whether our observations were related to a catchment scale impact or aquatic ecosystem impact we have performed and included the additional analyses suggested by the reviewer (see below). Overall, given our current data, we are in a better position to discuss drought impacts on aquatic ecosystems. Nevertheless, we have made an effort to consider whether these responses have catchment scale implications.

Does the composition of DOC inputs to these streams shift as a result of drought? I would expect the declining water tables would lead to more complete degradation of SOM and, perhaps, a reduction in complex DOC molecule export to streams... but perhaps the terrestrial and wetland components of these watersheds simply become disconnected by drought.

Response: Yes. Median DOC concentrations among KCS streams during summer 2018 was nearly half of that observed the previous summer (13 vs. 24 mg L⁻¹, n = 10; Fig. S6). These differences suggest that periods of low O₂ concentrations coincided with an overall isolation of streams from the shallow groundwater flow paths that deliver water and the bulk of DOC to these headwaters. In addition, absorbance data (i.e., SUVA) hint at the idea that DOM character also changed during drought, but these data are not nearly as clear. Such a change would not be that surprising given that the soil DOC sources likely shifted dramatically (e.g., from surficial organic soils to deeper mineral soils). Yet, we think that more sophisticated tools for DOM characterization would be needed to confirm this. See that this important point is now cover in the manuscript (lines 156 to 158 of revised manuscript).

What are the net GHG consequences of the decline in aerobic respiration and increase in anaerobic metabolism? Does drought increase or reduce freshwater GHG emissions when you calculate them in CO₂ equivalents. I think this is an important question for making this study of broader interest to ecosystem ecologists.

Response: We agree that the observed chemical and metabolic changes induced by drought might also translate into implications for GHG's balance in our stream network, an important point to broader the interest and impact of this study. To explore this, we have:

- 1) *estimated the daily CO₂ and CH₄ flux across the water-air interface (both in g CO_{2, eq} m⁻² d⁻¹) for each stream and date using Fick's First Law of gas diffusion*

(see additional information on the methodology used to estimate evasion fluxes in the Supplemental Information section; Methods S2).

- 2) *following the same approach used for Figure 6, we explored relationships between CO₂ and CH₄ emission fluxes and specific discharges for the five headwater streams in the KCS and during the summer season of the period between January 2010 and October 2018 (Fig.S8 and Table S4 of revised manuscript).*

Our initial prediction was that very low physical reaeration (measured as k_{600} ; in $m\ d^{-1}$) during drought would constrain gas evasion. Thus, CO₂ and CH₄ emission fluxes would increase with k_{600} from very low to median flow conditions. However, contrary to these expectations, both CO₂ and CH₄ emission fluxes remained relatively stable as discharge dropped during very low flow periods. CO₂ fluxes responded modestly to increasing runoff (i.e., more inclined slopes) compared to CH₄ fluxes. These observations suggest that the transition toward anaerobic over aerobic heterotrophic processes during drought (i.e., higher CH₄:CO₂ molar concentration ratios) is strong enough to overcome the effect of declining k_{600} on the CH₄ fluxes.

Following the suggestion of the reviewer and the results previously described, we have created a new paragraph to specifically discuss the net GHG consequences of observed biogeochemical responses to drought in the revised version (lines 193 to 213 of revised manuscript).

What if anything is known about how the catchment C balance responds to drought? How does this new information contribute to or change that understanding (either in terms of the spatial heterogeneity of responses OR in terms of the total magnitude of ecosystem change).

Response: This is a good question. Actually, we know that droughts increase the sink strength for CH₄ in terrestrial habitats both by limiting its production and increasing its oxidation when water tables are low (Fernner and Freeman, 2011; Strack et al., 2007), a pattern that has recently been confirmed for the 2018 summer drought in the Krycklan (Chi et al., 2019). We have expanded our discussion on the effects of droughts on the overall catchment C balance in northern regions, accounting for our new results in the stream, as well as the results from other publications in terrestrial ecosystems (lines 206 to 213 of revised manuscript).

Minor Comment

Lines 75-77 - I think there is a mistake here... did the WRT increase (as the #s suggest) or decrease (as the wording suggests). Spend a bit more time on this important point (since it is key to several of your graphs).

Response: Changed.

References

Chi, J., Nilsson, M., Laudon, H., Lindroth, A., Wallerman, J., Franssöm, J., et al. (2019). *The Net Landscape Carbon Balance – integrating terrestrial and aquatic carbon fluxes in a managed boreal forest landscape in Sweden. Global Change Biology.*

Reviewer #3 (Remarks to the Author):

This paper entitled "Drought-induced biogeochemical shifts in high latitude stream networks" by Gomez-Gener and others reports the effects of a manipulative drought experiment and a naturally dry year on aerobic and anaerobic biogeochemical processes. In both the manipulation and dry year, they found that in drought conditions, reduced compounds, including methane, were more abundant, correlated with water residence time. The study is unique in its spatial extent, demonstrating a network-wide response to drought that will be of broad interest to readers. Additionally, the coupling of a manipulation with a medium-term observational study gives added credibility to the conclusions. I have two minor concerns and several line edits, but after revision, I believe this study would be a valuable contribution to this journal.

Response: We thank reviewer 3 for the overall positive feedback provided to our study.

1. My main criticism is that the authors attribute all the changes to water residence time without exploring alternative hypotheses for the increase in reduced compounds. How much of the drought effects are due directly to hydrological changes versus other, indirect factors? For example, reaeration of oxygen varies with flow (more turbulence), and temperature is typically warmer during low flow conditions (Brown & Hannah 2007; Klaus et al. 2019). Because oxygen affect aerobic respiration and temperature regulates all the cited processes, drought it would substantially alter not only the hydrological residence time (which is the main dynamic described by the authors), it would alter the biogeochemical rates and the exposure time to conditions conducive for the different reactions (Oldham et al. 2013; Kolbe et al. 2019). I assume the manipulative experiment would have less variation in temperature than the drought year, but residence time is still tightly coupled to water flowpath and biogeochemical conditions (Abbott et al. 2016).

Response: We agree with the reviewer that alternative processes, all of them tightly coupled with hydrological variation, can also play a role in the described drought-induced biogeochemical patterns. In the revision, we have tried to give clearer balance to the hydrological and biological drivers of drought response in this landscape.

In terms of oxygen reaeration/resupply: decreased turbulence and water-atmosphere gas exchange is one of the factors restricting the re-supply of dissolved oxygen (O₂) to the stream. This factor depends on the hydrological change (the higher the discharge the higher the water-atmosphere gas exchange) and therefore follows the same direction as other hydrological-dependent factors limiting re-supply of O₂ (e.g., lateral or longitudinal vectors of O₂ transport). Therefore, the observed reduction of O₂ and increased rates of anaerobic processes in underlying sediments is controlled by hydrology, physics, and biology (aerobic respiration).

Specifically, the role of O₂ reaeration appears in the introduction of the current version of the manuscript (lines 51, 53 and 58 of revised manuscript). However, additional lines discussing the importance of this physical mechanism on driving the observed biogeochemical patterns during drought has been included in the revised version (lines

97 to 99 of revised manuscript). In addition, we agree that temperature can be an important driver of metabolic activity in streams. We therefore think that this factor likely had a positive effect on the processing of DOM, especially during the nature drought. Accordingly, we have included a sentence describing the pattern of temperature during the experiment (lines 89 to 91 of revised manuscript). However, we still propose that this temperature effect will most likely be overridden by drought-induced variations in water residence time: and associated limitation of the hydrological transport of organic substrates (e.g. DOM) and re-supply of O₂.

2. This kind of distributed, watershed-scale approach can reveal the extent of change, but can also be leveraged to identify the drivers of change (two recent examples from the Arctic: Connolly et al. 2018; Shogren et al. 2019). Besides looking at the distribution of oxygen concentrations with stream size, I wonder if the authors could explore how different catchment characteristics increase or decrease the likelihood of shifts into anoxia with drought. Said otherwise, could the authors use the rich spatiotemporal data from the Krycklan study area to explore why some rivers experienced persistent anoxia, while others were largely unaffected (e.g. explain the variability observed in Fig. 4)?

Response: We thank the reviewer for this important consideration and agree that it needs a more consideration. However, at the moment, we do not have a sufficient number of sites to do a robust (quantitative) analysis of how land cover predicts drought effects. Instead, we have added an additional paragraph about how drought is propagated in these heterogeneous landscapes, highlighting what we view as key considerations (e.g., variation in evapotranspiration and water storage) for understanding where and when such effects are likely to be most severe (lines 215 to 237 of revised manuscript).

Line edits:

Line 16: Because neither the experiment nor drought have been previously introduced, it is hard to parse this list.

Response: We have tried to better describe the multi-faceted approach used (see the abstract of revised manuscript).

44-59: This seems too detailed for this portion of the manuscript. Could a more general treatment of the relationship between residence time and biogeochemical conditions suffice here?

Response: We have left this text as is. In our view, this is the part of the Introduction where we connect the established models for metabolic processes under different redox conditions with some of the more unique properties of northern boreal catchments to set up our main hypothesis and predictions.

62: “Whether” and “if” are yes or no questions. “How much” and “to what degree” are richer and more informative.

Response: We thank the reviewer for this grammatical suggestion.

75: Typo (increased rather than reduced). Given the subject of the paper, perhaps avoid reduced altogether, except when referring to chemistry (decreased is clearer in this context).

Response: Changed.

77: How comparable is level of drought severity with amount of flow in the stream? The terrestrial environment, which sets the template for the water chemistry and which becomes more important if upstream flow is cut off as was done here, is not drought stressed, presumably.

Response: We agree with the reviewer that this definition can be confusing. Therefore, we have explicitly stated that the gradient of drought severity was observed for the stream reach but not the adjacent terrestrial ecosystem (lines 87 to 95 of revised manuscript).

Lines 82 and onward: The level of detail in the results feels a little imbalanced. Some of them are very specific to this study (PCA axes), while others are more applicable across studies (residence time versus redox state). Reorganizing and perhaps subdividing a little more could improve this.

Response: Done. We have tried to balance the level of detail across results. For the PCA case, we have substantially reduced the results and interpretation of results (lines 111 to 113 of revised manuscript).

119 and elsewhere: water chemistry is more specific than water quality

Response: We have changed this throughout the text. We only have kept the more general water quality in the abstract.

163-164: this seems like a valid hypothesis, but is it based on the results of the current study (as the sentence states) or the cited study?

Response: We agree with the reviewer that the cited study does not fit with the previous statement. We have thus removed it from the text.

166-167: Unclear. Potentially rephrase as “CH₄ production in aquatic environments could represent a larger proportion of ecosystem-level CH₄ balance during these years” or something like that.

Response: Changed (lines 193 to 213 of revised manuscript). Thanks for the suggested rephrasing.

168: typo “stimulates”

Response: Changed.

170: The primary product of denitrification, particularly in situations of long residence time, is the inert gas N₂. This can be an important ecosystem service in nutrient saturated environments, which could be emphasized as a positive tradeoff.

Response: Thanks for this suggestion. We have added a line including the potential for denitrification processes (line 116 of revised manuscript).

181-191: This seems largely redundant and could be removed without affecting the content of the paper.

Response: Thanks for the appreciation. However, we have opted to leave this concluding remark in the text.

Figure 4 is difficult to understand. Because it has different axes, the inset graph of the stream order might be more comprehensible as a separate panel.

Response: Thanks for this graphical suggestion. A new version with two panels has been included in the revised version of the manuscript.

REVIEWERS' COMMENTS:

Reviewer #1 (Remarks to the Author):

I am satisfied that the authors have carefully responded to and addressed my major concerns on the initial submission.

Reviewer #2 (Remarks to the Author):

I thank the authors for their serious attention to all reviewer suggestions and critiques. I thought the dataset was fantastic in round 1 and I now feel that the text is equally strong. The work is now more clearly contextualized and more cogently explained. I look forward to sharing this widely with colleagues.

Emily S. Bernhardt

Reviewer #3 (Remarks to the Author):

This revision of the manuscript entitled "Drought-induced biogeochemical responses in high latitude stream networks" by Gomez-Gener and others is greatly improved. The authors effectively integrated the comments from the three reviewers and the revised manuscript is much more convincing and polished. The added catchment information (vegetation type, surface-water and wetland coverage, etc.) give better context, the results are more logically organized, and the figures are both attractive and effective. I suggest a few line edits below, but otherwise consider the paper to be ready for publication.

Line 24: stream network (not streams networks)

Line 34: recent paper on land-use links with precipitation (Keys et al. 2019)

Lines 62-64: It seems like the mechanism is missing from this hypothesis. Is water residence the primary or sole process that is causing the shift? If so, I would mention it specifically. If not, listing a couple or few mechanisms linking drought with network-level chemistry would be helpful.

Line 215: All observations and projections of Arctic and Boreal runoff that I am aware of (from Peterson 2002 to many more recent publications) show an increase in annual river flow at high latitudes. The premise of this paper is that droughts may become more common. Are these two concepts in conflict, or could it be that river flow could increase at an annual timescale and drought also increases in frequency or severity?

Figure 5 b: The points overlap so much that they do not add much information to the panel. Could they be offset somewhat (jittered), made translucent, have a density trace added (violin plot), or be excluded (i.e. only show the statistical summary of the data)?

Citations:

Keys, P.W., Porkka, M., Wang-Erlandsson, L., Fetzer, I., Gleeson, T. & Gordon, L.J. (2019). Invisible water security: Moisture recycling and water resilience. *Water Secur.*, 8, 100046.

Peterson, B.J. (2002). Increasing River Discharge to the Arctic Ocean. *Science*, 298, 2171–2173.

We thank all three reviewers for comments and criticisms that improved this manuscript. In the section below, we provide point-by-point responses to the specific comments from the Reviewers based on the revised manuscript.

REVIEWERS' COMMENTS

A. Reviewer #1 (Remarks to the Author):

I am satisfied that the authors have carefully responded to and addressed my major concerns on the initial submission.

Author's response: We thank this reviewer for the constructive comments on the first round of revisions.

B. Reviewer #2 (Remarks to the Author):

I thank the authors for their serious attention to all reviewer suggestions and critiques. I thought the dataset was fantastic in round 1 and I now feel that the text is equally strong. The work is now more clearly contextualized and more cogently explained. I look forward to sharing this widely with colleagues.

Emily S. Bernhardt

Author's response: We thank this reviewer for the constructive comments on the first round of revisions.

C. Reviewer #3 (Remarks to the Author):

This revision of the manuscript entitled "Drought-induced biogeochemical responses in high latitude stream networks" by Gomez-Gener and others is greatly improved. The authors effectively integrated the comments from the three reviewers and the revised manuscript is much more convincing and polished. The added catchment information (vegetation type, surface-water and wetland coverage, etc.) give better context, the results are more logically organized, and the figures are both attractive and effective. I suggest a few line edits below, but otherwise consider the paper to be ready for publication.

Author's response: We appreciate the additional suggestions for improvement.

Line 24: stream network (not streams networks)

Author's response: We corrected this typo.

Line 34: recent paper on land-use links with precipitation (Keys et al. 2019)

Author's response: Thanks for bring this new paper to our attention. We have now added it as a reference.

Lines 62-64: It seems like the mechanism is missing from this hypothesis. Is water residence the primary or sole process that is causing the shift? If so, I would mention it specifically. If not, listing a couple or few mechanisms linking drought with network-level chemistry would be helpful.

Author's response: We made a small edit to this sentence to more clearly state that we consider water residence time (WRT) a primary factor causing biogeochemical shifts (lines 60 to 64 of revised manuscript). However, the point of these two sentences together is that, because of the high organic content in riparian soils of boreal headwaters, these streams may particularly sensitive increases in WRT. In other words, because organic matter supply to stream is so high, it might not take much change in the hydrology to drive the system toward more reducing conditions.

Line 215: All observations and projections of Arctic and Boreal runoff that I am aware of (from Peterson 2002 to many more recent publications) show an increase in annual river flow at high latitudes. The premise of this paper is that droughts may become more common. Are these two concepts in conflict, or could it be that river flow could increase at an annual timescale and drought also increases in frequency or severity?

Author's response: Thanks for this comment. The reviewer is correct: several studies document increases in river discharge from northern catchments, particularly for large Arctic rivers (Peterson et al. 2002). At the same time, there are also studies suggesting that drought events may become more common in some high latitude regions, including northern Scandinavia (e.g., Spinoni et al. 2018), and boreal Canada (e.g., Wang et al. 2014). While it is an interesting question, we are not in a position to speculate on whether these different trends are in conflict, nor is that the goal of our study. Instead, our data only address how headwater streams may respond to these events when they occur.

However, we have added to this revision references that explicitly provide future drought projections for northern Europe. For example, Spinone et al. 2018 not only identify northern Scandinavia as a region that may experience more frequent drought, but also highlight that these projections are variable space. In addition to this, we have revised the text (lines 242 to 246 of revised manuscript) to clarify that future drought projections for high latitudes are uncertain and likely to vary across northern landscapes based on a number of interacting drivers (e.g., temperature, precipitation) and catchment features (e.g., vegetation, permafrost storage, catchment water storage).

Figure 5 b: The points overlap so much that they do not add much information to the panel. Could they be offset somewhat (jittered), made translucent, have a density trace added (violin plot), or be excluded (i.e. only show the statistical summary of the data)?

Author's response: Thanks for this suggestion. Because the quantile regression lines provide information about the distribution of data for different stream orders, we have opted to leave this figure as is.

References:

- Keys, P. W., M. Porkka, L. Wang-Erlandsson, I. Fetzer, T. Gleeson, and L. J. Gordon. 2019. Invisible water security: Moisture recycling and water resilience. *Water Secur.* **8**: 100046. doi:10.1016/j.wasec.2019.100046
- Peterson, B. J., R. M. Holmes, J. W. McClelland, C. J. Vörösmarty, R. B. Lammers, A. I. Shiklomanov, I. A. Shiklomanov, and S. Rahmstorf. 2002. Increasing river discharge to the Arctic Ocean. *Science* (80-.). **298**: 2171–2173. doi:10.1126/science.1077445
- Spinoni, J., J. V. Vogt, G. Naumann, P. Barbosa, and A. Dosio. 2018. Will drought events become more frequent and severe in Europe? *Int. J. Climatol.* **38**: 1718–1736. doi:10.1002/joc.5291
- Wang, Y., E. H. Hogg, D. T. Price, J. Edwards, and T. Williamson. 2014. Past and projected future changes in moisture conditions in the Canadian boreal forest. *For. Chron.* **90**: 678–691. doi:10.5558/tfc2014-134